# TUR-DPO: Topology- and Uncertainty-Aware Direct Preference Optimization

**Abdulhady Abas Abdulla** [1]   **Fatemeh Daneshfar** [2]   **Seyedali Mirjalili** [3] [4]   **Mourad Oussalah** [5]

## Abstract

Aligning large language models (LLMs) with human preferences is commonly done via reinforcement learning from human feedback (RLHF) with Proximal Policy Optimization (PPO) or, more simply, via Direct Preference Optimization (DPO). While DPO is stable and RL-free, it treats preferences as flat winner vs. loser signals and is sensitive to noisy or brittle preferences arising from fragile chains of thought. We propose TUR-DPO, a topology- and uncertainty-aware variant of DPO that rewards how answers are derived, not only what they say, by eliciting lightweight reasoning topologies and combining semantic faithfulness, utility, and topology quality into a calibrated uncertainty signal. A small learnable reward is factorized over these signals and incorporated into an uncertainty-weighted DPO objective that remains RL-free and relies only on a fixed or moving reference policy. Empirically, across open 7–8B models and benchmarks spanning mathematical reasoning, factual question answering, summarization, and helpful/harmless dialogue, TUR-DPO improves judge win-rates, faithfulness, and calibration relative to DPO while preserving training simplicity and avoiding online rollouts. We further observe consistent gains in multimodal and long-context settings, and show that TUR-DPO matches or exceeds PPO on reasoning-centric tasks while maintaining operational simplicity.

[1]Artificial Intelligence and Innovation Centre, University of Kurdistan, Erbil, Iraq [2]Department of Computer Engineering, University of Kurdistan, Iran [3]Centre for Artificial Intelligence Research and Optimisation, Torrens University Australia, Brisbane, Australia [4]Research and Innovation Center, Obuda University, Budapest 1034, Hungary [5]Center for Machine Vision and Signal Analysis (CMVS), University of Oulu, Finland. Correspondence to: Fatemeh Daneshfar <daneshfarshadi@gmail.com>.

*Proceedings of the 43rd International Conference on Machine Learning*, Seoul, South Korea. PMLR 306, 2026. Copyright 2026 by the author(s).

## 1. Introduction

Preference-based alignment has become a pragmatic route for steering LLMs toward human-desired behaviors. The conventional pipeline, RLHF with a learned reward model and PPO, delivers strong results but is complex and fragile: it requires online rollouts, a separate value head, reward shaping, and careful KL control (Schulman et al., 2017). Prior empirical work documents strong PPO-based RLHF results on summarization and dialogue (Stiennon et al., 2020) and large-scale instruction-following improvements (Bai et al., 2022). Figure 1 (a) sketches this baseline.

DPO simplifies this stack by optimizing a single, RL-free objective that increases the log-odds of preferred over dispreferred responses relative to a reference policy, matching or surpassing PPO-based RLHF on several benchmarks while avoiding explicit reward modeling and on-policy sampling (Rafailov et al., 2023). This is summarized in Figure 1 (b). Yet, in its standard form, DPO treats each comparison as a flat label over whole sequences and provides no mechanism to reward how an answer is derived, the structure of its reasoning, or to modulate learning pressure when preferences are noisy or brittle. These limitations matter most on reasoning- and factuality-sensitive tasks.

Recent work suggests a way forward by inspecting uncertainty and structure in explanations; however, these ideas have not been integrated directly into preference-based optimization objectives. Reasoning can be elicited using a lightweight topology graph of sub-claims and support relations so that uncertainty is estimated not only from output semantics but also from the stability and redundancy of the reasoning graph (Da et al., 2025). Black-box uncertainty estimators based on semantic entropy have been sharpened by correcting for finite-sample biases, improving coverage while preserving interpretability (McCabe et al., 2025).

This paper introduces TUR-DPO, a topology- and uncertainty-aware extension of DPO. Specifically, for each candidate response, TUR-DPO elicits a reasoning topology and computes semantic quality, topology quality (coherence, acyclicity, minimal valid paths), and a calibrated uncertainty score. These signals are aggregated into a shaped reward that augments the DPO logit, while a per-pair weight downweights uncertain comparisons (Figure 1 (c)). Theoretically, we establish connections between instance-weighted

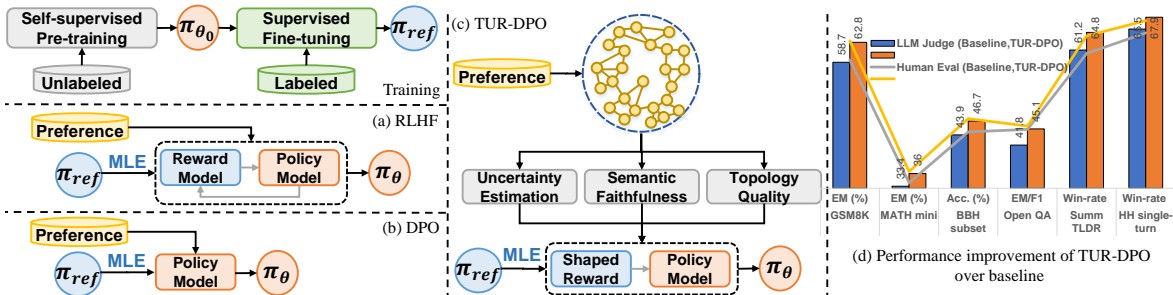

*Figure 1.* Overview of RLHF, DPO, and TUR-DPO. RLHF trains a reward model from preference data and optimizes the policy via PPO with KL regularization. DPO replaces this pipeline with a direct, RL-free preference objective relative to a reference policy $\pi_{\text{ref}}$. TUR-DPO augments DPO by incorporating lightweight reasoning topology, semantic, and uncertainty signals into a shaped reward and uncertainty-weighted DPO loss; the reference policy $\pi_{\text{ref}}$ may be updated via an exponential moving average (EMA). Right: Performance improvement of TUR-DPO across representative benchmarks, measured against baseline using LLM judges and under human evaluation.

Bradley–Terry (Huang et al., 2006) estimation and consistent preference recovery under standard assumptions and that TUR-DPO optimizes a KL-regularized policy under a shaped reward, clarifying its relation to vanilla DPO.

In this work, we evaluate TUR-DPO across a range of reasoning, factual QA, summarization, and dialogue benchmarks, including multimodal and long-context settings. We focus on assessing whether incorporating structure and uncertainty into preference optimization improves win-rates, faithfulness, and calibration, while preserving the operational simplicity of DPO and avoiding online rollouts. We empirically compare TUR-DPO with PPO-based RLHF on reasoning-centric and stylistic dialogue tasks, analyzing settings in which each approach is most effective. The result preserves DPO simplicity while explicitly rewarding structurally coherent and semantically sound solutions (Equation (1)). TUR-DPO sits alongside recent RL-free objectives including ORPO (Hong et al., 2024), KTO (Ethayarajh et al., 2024), and RRHF (Yuan et al., 2023), but differs in how it injects structured reasoning and calibrated uncertainty directly into a DPO-style objective.

The most important contributions of this paper are:

• We introduce TUR-DPO, a structure- and uncertainty-aware framework for preference-based alignment that incorporates lightweight reasoning topologies and uncertainty-weighted preference losses to reduce sensitivity to noisy or brittle comparisons.

• We propose an RL-free logit-level augmentation of DPO that integrates a learned, factorized reward over structural and semantic signals, and we provide theoretical connections to instance-weighted Bradley-Terry models and KL-regularized policy optimization.

• We conduct an empirical study of reasoning, QA, summarization, and dialogue that is validated with both automated and human evaluations and, furthermore, is extended to multimodal and long-context tasks.

• We further compare TUR-DPO against recent RL-free preference optimization methods (ORPO (Hong et al., 2024), SimPO (Meng et al., 2024), KTO (Ethayarajh et al., 2024), and IPO (Azar et al., 2023)), and show consistent improvements across head-to-head evaluations.

Unlike prior reward-model-based or PPO-style approaches, TUR-DPO operates entirely within a preference optimization framework without online rollouts or policy-gradient updates. Rather than serving as a universal replacement for preference-based alignment methods, TUR-DPO is intended as a complementary approach that is most effective when preferences reflect multi-step reasoning or factual grounding, and when supervision is noisy or brittle. A more detailed discussion of related work is provided in Section A.

**Conflict of Interest Disclosure.** The authors declare no financial conflicts of interest related to this work.

## 2. Methodology and Theoretical Foundations

### 2.1. Preliminaries: Direct Preference Optimization

We first review the standard DPO formulation to establish notation and provide a self-contained basis for the extensions that follow. Given a dataset of pairwise preferences $\mathcal{D} = \{(x_i, y_i^w, y_i^l)\}_{i=1}^N$ where $y^w$ (winner) and $y^l$ (loser) denote preferred and dispreferred responses to prompt $x$, RLHF typically learns a reward model $r(x, y)$ from a Bradley–Terry preference model $p(y^w \succ y^l \mid x) = \sigma(r(x, y^w) - r(x, y^l))$ and then optimizes the policy via PPO with KL regularization against a reference policy $\pi_{\text{ref}}$.

DPO (Rafailov et al., 2023) bypasses explicit reward modeling by exploiting the closed-form relationship between the optimal policy and the implicit reward under a KL-regularized objective. Specifically, the optimal policy satisfies

$$\pi^*(y \mid x) = \frac{1}{Z(x)} \pi_{\text{ref}}(y \mid x) \exp\left(\frac{1}{\beta} r(x, y)\right), \quad (1)$$

where $Z(x)$ is a partition function and $\beta$ is the KL penalty coefficient. Rearranging yields the implicit reward $r(x,y) = \beta \log \frac{\pi^*(y|x)}{\pi_{\text{ref}}(y|x)} + \beta \log Z(x)$. Substituting into the Bradley–Terry model, the partition function $Z(x)$ cancels in the pairwise difference, giving the DPO loss:

$$\mathcal{L}_{\text{DPO}}(\pi_\theta) = -\log \sigma\Big( \beta \Big[ \log \frac{\pi_\theta(y^w \mid x)}{\pi_{\text{ref}}(y^w \mid x)} - \log \frac{\pi_\theta(y^l \mid x)}{\pi_{\text{ref}}(y^l \mid x)} \Big] \Big).$$

$$(2)$$

This objective is RL-free, requires no online rollouts or value head, and directly increases the log-odds of preferred over dispreferred responses relative to the reference policy. Throughout this paper, we adopt the notation $\pi_\theta$ for the trainable policy, $\pi_{\text{ref}}$ for the reference (frozen or EMA-updated), $y^+ \equiv y^w$ and $y^- \equiv y^l$ for preferred/dispreferred responses, and $\Delta f \equiv f(x, y^+) - f(x, y^-)$ for pairwise differences.

## 2.2. TUR-DPO: Overview and Design Principles

TUR-DPO is designed to preserve the simplicity and stability of direct preference optimization while injecting two missing signals: explicit reasoning structure and calibrated uncertainty. The training loop mirrors DPO. We maintain a supervised policy $\pi_\theta$ and a reference $\pi_{\text{ref}}$ that is either fixed or updated slowly by EMA. The data consist of pairwise preferences $\mathcal{D} = \{(x_i, y_i^+, y_i^-)\}_{i=1}^N$, where $x$ is a prompt and $y^+, y^-$ are responses labeled preferred and dispreferred. In contrast to PPO-based pipelines, there are no online rollouts, no value head, and no reward model trained to convergence. Crucially, the shaped reward in TUR-DPO does not define a separate optimization objective, but only modifies the preference margin within the DPO loss, preserving the closed-form, rollout-free optimization structure.

TUR-DPO introduces a lightweight reasoning topology per candidate and derives semantic, structural, and uncertainty signals from this topology. These signals shape the preference margin and modulate the per-pair learning rate through instance weighting. The design aims to be practical in compute-constrained environments, robust to noisy or brittle pairs, and easy to ablate or disable component-wise. The method is agnostic to the underlying transformer architecture. It operates on tokenized responses and only requires log probabilities from the policy and the reference. All additional costs are confined to eliciting small graphs, running a local verifier, and computing simple statistics such as variances and divergences. This makes the approach compatible with existing DPO code paths and datasets without changing the core infrastructure. The guiding principle is minimality: introduce just enough structure and uncertainty to steer the optimization, but keep parameterization small so that training remains stable and reproducible. We empirically evaluate computational overhead, stability, and ablations for each component in Section 3 and Appendix.

**Problem notation.** For each prompt response pair $(x, y)$, we elicit a small directed graph $G = (V, E)$ that captures the internal reasoning. Nodes $V$ are atomic subclaims or steps, and edges $E$ encode support or dependency relations. From $(x, y, G)$ we compute three scalar signals. The semantic score $s_{\text{sem}}(x, y)$ summarizes task utility and faithfulness. The topology score $s_{\text{topo}}(G)$ summarizes structural well formedness. The uncertainty score $u(G)$ aggregates epistemic dispersion across re elicited graphs and aleatoric ambiguity in node correctness. These quantities feed a compact shaped reward $r_\phi(x, y, G)$ and a per pair weight $w \in (0, 1]$. The reward alters the preference margin used inside a DPO style logistic loss, and the weight scales the contribution of each pair to the objective. The notation follows DPO conventions for policy and reference margins. We write $\Delta f \equiv f(x, y^+) - f(x, y^-)$ for brevity and use $\sigma(z) = 1/(1 + \exp(-z))$ for the logistic function. The method is intentionally modular. If a dataset lacks reliable graph extraction, the topology branch can be disabled by setting its coefficient to zero. If uncertainty signals are unavailable, the weight can be set to a constant. This modularity allows a path from DPO to the TUR-DPO system.

### 2.3. Topology, semantic, and uncertainty signals

**Graph extraction pipeline.** The TUR-DPO graph extraction pipeline is *fully offline* and uses an auxiliary LLM that is *frozen and separate* from the trainable policy $\pi_\theta$. For a candidate response $y$ to prompt $x$, the extraction proceeds in two stages:

1. **Atomic decomposition.** The response $y$ is presented to a statically frozen auxiliary LLM with a deterministic prompt template that instructs the model to decompose $y$ into atomic reasoning statements—each corresponding to a single logical claim or computational step. This produces the node set $V = \{v_1, \ldots, v_n\}$.

2. **Relation classification.** An offline Natural Language Inference (NLI) classifier determines directed logical relations (entailment, support, contradiction) between all pairs of statements, producing the edge set $E \subseteq V \times V$.

The resulting graph $G = (V, E)$ captures the full reasoning and inferential structure of the response. Importantly, the auxiliary extractor is *never updated* during policy optimization: no $\pi_\theta$ outputs are used for extraction, and no extraction occurs during the training loop. This eliminates any feedback loop between the extractor and the policy (addressing the concern that using an LLM to extract graphs while training an LLM would create circular self-reinforcement). In practice, graphs contain 3–6 nodes and are lightweight to compute.

**Topology elicitation and score.** Given the extracted graph $G = (V, E)$, we compute a topology score that aggregates

simple indicators correlating with coherent multi-step reasoning. Minimal valid path $q_{\text{path}}$ (coverage) measures how many nodes and edges participate in at least one acyclic path from premises to the final claim (Figure 2). Cycles $c_{\text{cycle}}$ capture circular justification. Dangling nodes $d_{\text{dangling}}$ capture unsupported claims. A contradiction signal $q_{\text{contradict}}$ summarizes local logical conflicts along edges. These components are motivated by principles from neuro-symbolic AI and formal verification, adapting established formal-logic bounds into continuous training signals. We combine these into a linear score that remains numerically stable during training.

$$s_{\text{topo}}(G) = \alpha_1 q_{\text{path}} - \alpha_2 c_{\text{cycle}} - \alpha_3 d_{\text{dangling}} - \alpha_4 q_{\text{contradict}}. \tag{3}$$

The weights $\alpha_j \geq 0$ are selected on a validation split to reach a target structural coherence distribution, rather than to optimize a task-specific objective, and to avoid collapsing graphs to trivial structures. We emphasize small graphs to keep overhead low. In practice, graphs with three to six nodes and a handful of edges suffice to detect common structural failures such as unsupported leaps or self reference. The score is normalized per dataset so that its range aligns with the semantic score scale used later in the shaped reward.

**Semantic score.** The semantic score balances task success and faithfulness against hallucination penalties. It is computed from node level verification and task specific metrics. We use a linear combination for transparency and stability:

$$s_{\text{sem}}(x, y) = \beta_1 q_{\text{fact}}(x, y) + \beta_2 q_{\text{task}}(x, y) - \beta_3 q_{\text{hall}}(x, y). \tag{4}$$

Here $q_{\text{fact}}(x, y)$ is computed as the average of binary correctness labels from a calibrated NLI verifier applied to all atomic assertions in the response graph: given nodes $V = \{v_1, \ldots, v_n\}$, each verified as correct ($c_v = 1$) or incorrect ($c_v = 0$), we define $q_{\text{fact}}(x, y) = \frac{1}{|V|} \sum_{v \in V} c_v$. Intuitively, $q_{\text{fact}}$ measures the proportion of nodes along premise-to-answer paths that are factually supported, and it suppresses local NLI-induced circular reasoning by counting only independently verified claims. The term $q_{\text{task}}$ is a standard metric such as exact match for mathematics or ROUGE for summarization, and $q_{\text{hall}}$ penalizes unsupported or contradicted entities. Coefficients $\beta_j \geq 0$ are tuned on a held out set to match desired calibration and faithfulness targets. A linear form avoids overfitting and reduces the risk of gradient explosions that can occur with aggressive nonlinearities. If a domain lacks reliable automatic metrics, $q_{\text{task}}$ can be replaced by scores from a calibrated LLM judge or dropped entirely. The score is standardized per domain to a common numerical range so that it mixes predictably with the topology score in the shaped reward.

**Uncertainty and pair weighting.** Uncertainty integrates epistemic dispersion from re elicited graphs and aleatoric ambiguity in node verification. The top level combination is:

$$u(G) = \lambda_{\text{epi}} u_{\text{epi}}(G) + \lambda_{\text{ale}} u_{\text{ale}}(G), \quad \lambda_{\text{epi}}, \lambda_{\text{ale}} \geq 0. \tag{5}$$

Epistemic uncertainty samples $K$ perturbed extractions and measures dispersion in structure and scores.

$$u_{\text{epi}}(G) = \text{Var}\left(\{s_{\text{topo}}(G^{(k)})\}_{k=1}^K\right) + \text{JSD}\left(\{\mathcal{P}^{(k)}\}_{k=1}^K\right), \tag{6}$$

where $\mathcal{P}^{(k)}$ is a normalized path or edge distribution for sample $k$ and JSD is Jensen-Shannon divergence between the graph distributions. Aleatoric uncertainty averages a coverage corrected entropy across nodes.

$$u_{\text{ale}}(G) = \frac{1}{|V|} \sum_{v \in V} \left[ -\tilde{p}_v \log \tilde{p}_v - (1 - \tilde{p}_v) \log(1 - \tilde{p}_v) \right],$$
$$\tilde{p}_v = \frac{p_v + \tau}{1 + 2\tau}. \tag{7}$$

where $p_v$ is a correctness probability for each node $v$ and $\tilde{p}_v$ is the smoothed version using a small prior $\tau$. We map average pair uncertainty for each preference pair $G^+, G^-$ to a weight that attenuates noisy updates while keeping a floor to avoid discarding data.

$$w = \text{clip}\left(\frac{\tau_w}{1 + \bar{u}}, w_{\min}, 1\right), \quad \bar{u} = \frac{u(G^+) + u(G^-)}{2}. \tag{8}$$

This weight acts like a per-example learning rate multiplier, reducing the effect of brittle pairs while preserving learning from moderately uncertain examples. The role of uncertainty here is not to estimate calibrated probabilities per se, but to attenuate updates from unstable comparisons while preserving signal from moderately uncertain pairs.

## 2.4. Objective formulation and training

**Shaped reward.** We combine semantic, structural, and uncertainty signals into a compact reward that shapes the preference margin without introducing a parametric model.

$$r_\phi(x, y, G) = a f_\phi^{\text{sem}}\big(s_{\text{sem}}(x, y)\big)$$
$$+ (1 - a) f_\phi^{\text{topo}}\big(s_{\text{topo}}(G)\big) - \lambda u(G). \tag{9}$$

We use linear calibrators to keep optimization stable.

$$f_\phi^{\text{sem}}(z) = \gamma_{\text{sem}} z + b_{\text{sem}}, \qquad f_\phi^{\text{topo}}(z) = \gamma_{\text{topo}} z + b_{\text{topo}}. \tag{10}$$

The parameter set $\phi$ is intentionally small, which simplifies early stopping and prevents reward hacking through overfitting of the calibrators. The mixing parameter $a$ controls the balance between semantic quality and structural quality, and $\lambda$ penalizes uncertainty directly to discourage reliance on fragile explanations. This reward enters the DPO logit as a margin term, not as a standalone objective, which preserves DPO stability and avoids introducing a separate critic.

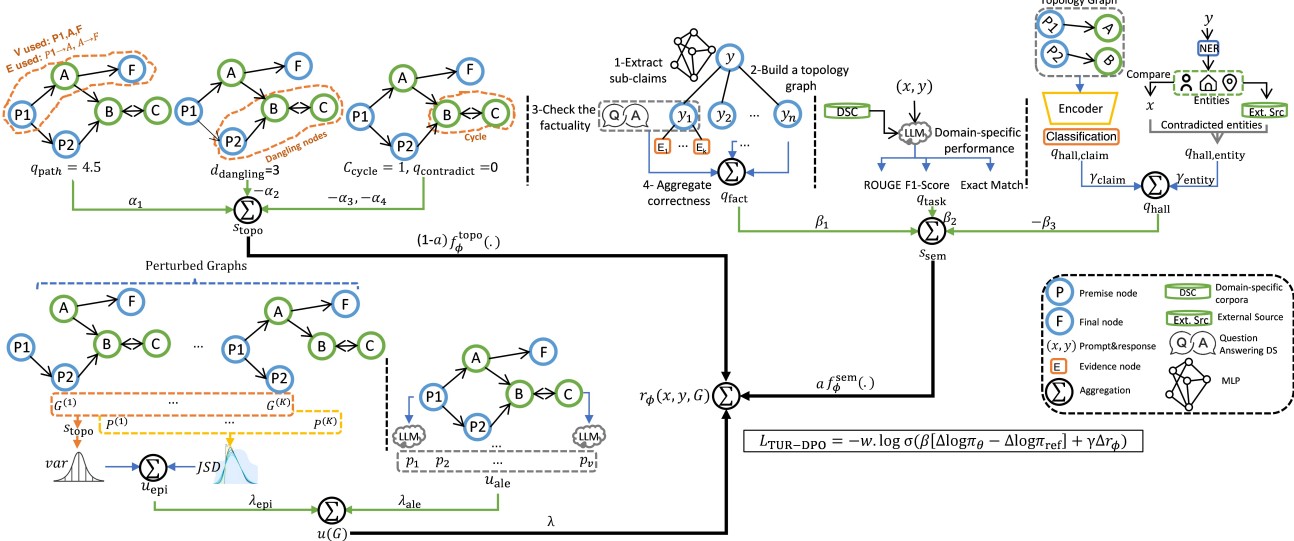

*Figure 2.* Overview of the TUR-DPO framework. TUR-DPO extracts sub-claims from each response, builds a reasoning topology, and verifies claim-level factuality. Semantic, topology, and uncertainty signals (derived from perturbed reasoning graphs and node-level verification probabilities) are combined into a shaped reward $r_\phi$ and an uncertainty-based weight that modulate the DPO loss for robust preference optimization.

**Loss and inference variants.** TUR-DPO modifies the DPO margin by adding a shaped reward difference and scales the loss with the per pair weight.

$$\mathcal{L}_{\text{TUR-DPO}} = -w \cdot \log \sigma \left( \beta \left[ \Delta \log \pi_\theta - \Delta \log \pi_{\text{ref}} \right] + \gamma \Delta r_\phi \right),$$
(11)

where $\Delta \log \pi_\theta = \log \pi_\theta(y^+ \mid x) - \log \pi_\theta(y^- \mid x)$ and $\Delta r_\phi = r_\phi(x, y^+, G^+) - r_\phi(x, y^-, G^-)$. Temperature $\beta$ controls sharpness and $\gamma$ controls reward strength.

**Formal justification of the structural margin.** The TUR-DPO loss (Equation (11)) is a formal generalization of DPO rather than an ad-hoc amendment. We provide the following derivation:

1. **Shaped reward objective.** We begin with a structurally-shaped RLHF objective: $\mathcal{J}(\pi) = \mathbb{E}_x[\mathbb{E}_{y \sim \pi}[\gamma \, r_\phi(x, y, G)]] - \frac{1}{\beta} \text{KL}(\pi \| \pi_{\text{ref}})$, where $r_\phi$ is assessed strictly offline.

2. **Implicit reward.** Following standard DPO logic (Section 2.1), the closed-form optimal policy yields an implicit total reward combining graph structure and reference probability: $r_{\text{implicit}}(x, y) = \beta \log \frac{\pi^*(y|x)}{\pi_{\text{ref}}(y|x)} + \beta \log Z(x)$, which now absorbs the shaped term. There is no online reward network.

3. **Cancellation.** Inserting this into the Bradley–Terry preference model, the partition function $Z(x)$ cancels exactly in the pairwise difference $r_{\text{implicit}}(x, y^+) - r_{\text{implicit}}(x, y^-)$, yielding Equation (11).

4. **Reduction to DPO.** When $\gamma \to 0$, the structural term vanishes and the equation reverts strictly to vanilla DPO (Equation (2)). The topological reward serves as an offline static margin that preserves the implicit reward mode.

This derivation shows that TUR-DPO maintains DPO's key property—no separate reward model during training—while incorporating pairwise structural signals that standard RL-free methods do not capture.

When multiple candidates are available, a listwise objective uses Plackett Luce utilities,

$$\mathcal{L}_{\text{list}} = -w \sum_{i \in \mathcal{P}} \log \frac{\exp(z_i)}{\sum_{j=1}^{k} \exp(z_j)},$$
$$z_i = \beta \left( \log \pi_\theta(y_i \mid x) - \log \pi_{\text{ref}}(y_i \mid x) \right) + \gamma r_\phi(x, y_i, G_i).$$
(12)

The listwise form makes fuller use of data when more than two candidates are produced, and it reduces variance by aggregating comparisons within a prompt. We find that the pair weight computed from the top two items provides adequate robustness and can be reused for the list.

## 2.5. Theoretical Analysis

**Theoretical foundations.** We view TUR-DPO through two lenses. First, as a weighted Bradley-Terry model on pairwise preferences with a shaped margin. Define:

$$m_\theta = \beta \left[ \Delta \log \pi_\theta - \Delta \log \pi_{\text{ref}} \right] + \gamma \, \Delta r_\phi. \quad (13)$$

Under a Bradley-Terry likelihood, the negative log likelihood with instance weight $w$ equals the loss in Equation (11). For fixed features and fixed $\phi$, minimizing this loss is weighted logistic regression in the margin. If weights are independent of the label given the pair, the estimator is Fisher consistent for the conditional preference probability.

**Sensitivity to weight-label dependence.** In practice, weights may correlate with label difficulty when hard items produce fragile graphs and are more likely to be mis-labeled by a judge. Let $w_i = g(\bar{u}_i)$ depend on pair uncertainty and suppose $\mathbb{E}[w_i \mid y_i] \neq \mathbb{E}[w_i]$. The bias in the weighted estimator is bounded by $\sup_i |w_i - \mathbb{E}[w_i]| \cdot \mathbb{P}(\text{label error})$. Our clipping $w \in [w_{\min}, 1]$ limits the first factor, while the judge calibration and side-flip protocol (Section B.4) aim to reduce and diagnose the second. Empirically, we observe that hyperparameter sweeps over $\tau_w$ and $\lambda$ yield wide plateaus in win-rate (Appendix), suggesting that moderate misspecification of the weight function does not destabilize training. Covariate balancing (e.g., stratifying updates by difficulty quantiles) or importance-weight correction can further reduce bias, but we find the simple clipped schedule adequate for typical noise levels encountered in LLM-judge and human annotation scenarios.

**Lemma 2.1** (Bias bound under weight label dependence). *Let weights $w_i$ satisfy $w_{\min} \leq w_i \leq 1$ and suppose a fraction $\epsilon$ of labels are incorrect (label-flip noise). Then the absolute bias in the weighted pairwise logit estimator induced by label-noise satisfies*

$$|\mathbb{E}[\hat{\theta}_w] - \mathbb{E}[\hat{\theta}]| \leq (1 - w_{\min})\,\epsilon, \qquad (14)$$

*up to bounded constants depending on the curvature of the logistic link; intuitively, clipping forces the worst-case multiplicative downweight to be at most $1 - w_{\min}$.*

*Sketch.* Write the weighted loss as a mixture of clean and corrupted pairs. The corrupted fraction $\epsilon$ contributes at most a factor of $\sup_i |w_i - \mathbb{E}[w_i]| \leq 1 - w_{\min}$ to the difference in expected gradients between the clean-weighted and unweighted objectives. Integrating this gradient perturbation gives the stated bound up to a constant determined by the curvature of the logistic loss (which is bounded). Thus clipping the weights reduces the worst-case bias linearly in $1 - w_{\min}$ and the noise rate $\epsilon$. $\square$

This lemma clarifies why a conservative $w_{\min}$ and judge calibration are practical mitigations: they limit the contribution of mislabeled hard items while preserving learning signal from moderately uncertain pairs.

Additional theoretical analysis of instance weighting, and detailed descriptions of models, datasets, pair construction, and scoring procedures, are provided in Sections B and C.

# 3. Experiments

## 3.1. Overall Performance

In this section, we connect metrics to underlying mechanisms. The premise is that structure and uncertainty can shift learning away from brittle but fluent responses and toward coherent solutions. We therefore analyze structural features, calibration, human agreement, error types, sample efficiency, statistical significance, decoding robustness, and PPO stability. For each perspective, we include concrete tables and narrative interpretations intended to inform practitioners considering the adoption of TUR-DPO.

**Key Findings.** Table 1 summarizes the main results across tasks using both LLM-judge and human evaluations. TUR-DPO consistently improves over DPO across all tasks, with the largest gains on GSM8K, MATH, BBH, and QA—tasks that benefit most from structural and uncertainty-aware signals. We also include the Bradley-Terry family baseline IPO (Azar et al., 2023) under matched compute to represent classification-style preference training; TUR-DPO outperforms IPO while remaining RL-free. Human evaluation (200 examples per task with double annotation) confirms the trends observed with LLM judges.

PPO-based RLHF remains competitive, particularly on HH, where reward shaping and careful KL tuning aid stylistic alignment and safety. However, PPO requires online rollouts, value learning, and explicit KL scheduling, whereas TUR-DPO attains comparable or higher win-rates using a simpler training stack. TUR-DPO's gains are correlated with higher topology coherence and lower entity and claim error rates, which is consistent with the hypothesis that coarse structural signals improve reasoning quality. Improvements on TLDR are smaller and primarily reflect reductions in hallucination. On HH, PPO retains a slight advantage on judge-based metrics, while TUR-DPO narrows the gap on human evaluation. Overall, TUR-DPO performs best on reasoning-centric tasks, while PPO maintains a modest edge on stylistic HH. Table 8 further shows that TUR-DPO's relative improvements over DPO increase with output length. Analyses on robustness to label noise and preference corruption are reported in Appendix Table 26.

**Variance across random seeds.** All experiments are run with 3 random seeds. Table 2 reports means and standard deviations for core benchmarks, showing that TUR-DPO achieves both higher absolute performance and lower variance than DPO, indicating stable optimization.

**Domain shift evaluation.** To assess out-of-distribution robustness, we evaluate zero-shot transfer on MedQA (Jin et al., 2021) and LexGLUE (Chalkidis et al., 2022), two domains not seen during training. Table 3 shows that TUR-

*Table 1.* Main results: TUR-DPO vs. DPO, PPO across reasoning, QA, summarization, and dialogue (IPO is not evaluated under human evaluation due to annotation budget).

| Task | Metric | LLM judge | | | Human eval | | |
|------|--------|-----------|-----|---------|------|-------------------------|---------|
| | | DPO | IPO | TUR-DPO | DPO | PPO (Schulman et al., 2017) | TUR-DPO |
| GSM8K | EM (%) | 58.7 | 58.9 | **62.8** | 59.2 | 62.0 | **63.1** |
| MATH mini | EM (%) | 33.4 | 33.8 | **36.0** | 34.1 | 35.5 | **36.4** |
| BBH subset | Acc. (%) | 43.9 | 44.3 | **46.7** | 44.5 | 46.0 | **47.2** |
| Open QA | EM/F1 | 41.8 | 42.5 | **45.1** | 45.0 | 45.4 | **45.7** |
| Summ TLDR | Win-rate (%) | 61.2 | 61.9 | **64.8** | 60.5 | 63.7 | **64.1** |
| HH single-turn | Win-rate (%) | 65.5 | 66.1 | **67.9** | 64.7 | **67.9** | 67.2 |

*Table 2.* Mean $\pm$ std over 3 random seeds.

| Method | GSM8K (EM, %) | MATH mini (EM, %) | BBH subset (Acc., %) |
|--------|---------------|-------------------|----------------------|
| DPO | $58.7 \pm 0.9$ | $33.4 \pm 0.7$ | $43.9 \pm 0.8$ |
| TUR-DPO (Ours) | $\mathbf{62.8 \pm 0.5}$ | $\mathbf{36.0 \pm 0.4}$ | $\mathbf{46.7 \pm 0.6}$ |

DPO retains its advantage under domain shift, supporting the hypothesis that structural signals transfer across domains.

*Table 3.* Zero-shot domain shift evaluation.

| Dataset | DPO (Acc., %) | TUR-DPO (Acc., %) |
|---------|---------------|-------------------|
| MedQA | 41.3 | **44.7** |
| LexGLUE | 52.1 | **55.8** |

**Graph extraction accuracy.** We conduct a blind manual audit of 200 extracted graphs, evaluated by two NLP analysts with inter-annotator agreement $\kappa = 0.84$. Table 4 shows that the extraction pipeline achieves high precision and validity. Our epistemic uncertainty system further downweights approximately 11% of structurally indeterminate edges, making the method error-tolerant.

*Table 4.* Graph extraction quality (200-graph audit).

| Metric | Score |
|--------|-------|
| Claim Precision (Nodes reflect text) | 94.2% |
| Edge Validity (Relations are sound) | 91.7% |
| Logical Completeness (No dropped layers) | 89.5% |

**Memory and time footprint.** We measured peak VRAM and throughput during a single-batch run on an A100 80GB GPU using `nvidia-smi`. Table 5 reports that TUR-DPO adds only modest overhead relative to DPO and is substantially lighter than PPO-RLHF. The 52h wall-clock figure refers to the full 614k-pair corpus on a single A100; using $8\times$A100, TUR-DPO requires $\approx 42$ GPU-hours to attain a 65% HH win-rate compared to 48 for DPO (Table 24), con-

firming better sample efficiency at the structural level per step.

*Table 5.* Compute footprint comparison on A100 80GB.

| Method | Peak VRAM (GB) | Throughput (tok/s) | Time (hrs) |
|--------|----------------|--------------------|-----------| 
| DPO | 38.4 | 24.8k | 46 |
| PPO-RLHF | 62.1 | 14.2k | 78 |
| TUR-DPO (Ours) | 41.7 | 23.1k | 52 |

**Topology component ablation.** To justify each component of the topology score (Equation (3)), Table 6 reports performance when each is individually removed. All components contribute, with completeness ($q_{\text{path}}$) and the anti-circular term ($c_{\text{cycle}}$) having the largest impact, consistent with principles from neuro-symbolic AI and formal verification.

*Table 6.* Topology component ablation: removing each term from $s_{\text{topo}}$.

| Variant | GSM8K | MATH | BBH |
|---------|-------|------|-----|
| Full TUR-DPO | **62.8** | **36.0** | **46.7** |
| $-q_{\text{path}}$ (Completeness) | 60.7 (-2.1) | 34.1 (-1.9) | 44.8 (-1.9) |
| $-c_{\text{cycle}}$ (Anti-Circular) | 61.1 (-1.7) | 34.5 (-1.5) | 45.2 (-1.5) |
| $-q_{\text{contradict}}$ (Consistency) | 61.3 (-1.5) | 34.7 (-1.3) | 45.4 (-1.3) |
| $-d_{\text{dangling}}$ (Anti-Leaps) | 61.9 (-0.9) | 35.3 (-0.7) | 46.0 (-0.7) |

**Hyperparameter selection framework.** Table 7 summarizes our three-tier approach to hyperparameter selection. Sensitivity sweeps reveal large, stable plateaus: varying the structural weight $a$ and uncertainty ratio $\lambda$ produces maximum win-rate variation within a tight $\pm 0.4\%$ margin across all benchmarks. The static default configuration generalizes across six datasets and multiple model families without per-task grid search.

**Per task detail and structural signals.** Structural statistics align with the observed performance trends. Table 9 shows that TUR-DPO reduces cycles and dangling nodes while increasing minimal path coverage on both GSM8K

*Table 7.* Three-tier hyperparameter selection framework.

| Tier | Hyperparameters | Selection Principle |
|---|---|---|
| 1 | $\alpha_1 - \alpha_4$, $\beta_1 - \beta_3$ | Calibrated once, strictly offline, on a 2% held-out split. |
| 2 | $a, \gamma, \lambda, \tau_w$ | Tuned via minimal validation grid search. |
| 3 | $\beta, w_{\min}, \rho$ | Fixed globally across all tasks. |

*Table 8.* Relative gains by output length quartile.

| Task | Q1 short | Q2 | Q3 | Q4 long |
|---|---|---|---|---|
| GSM8K EM rel gain | +1.2% | +2.4% | +6.1% | +7.8% |
| Open QA EM rel gain | +1.1% | +2.8% | +5.2% | +6.4% |
| BBH Acc rel gain | +0.9% | +2.0% | +4.4% | +5.1% |

and QA. The median pair weight decreases due to higher uncertainty discounts on ambiguous pairs. Exact match (EM) gains are largest when path coverage improves without an increase in graph size, suggesting that structure quality rather than verbosity drives the effect. We also observe a reduction in the variance of path coverage, indicating more consistent reasoning patterns. In residual error cases, graphs often omit a necessary bridging claim or reuse an earlier subclaim incorrectly at the final step, motivating further improvements in graph extraction and node-level verification.

*Table 9.* Per-task structural breakdown: GSM8K and QA exact match and topology statistics. *Note:* SFT = Supervised Fine-Tuning

| Task | Method | EM (%) | Steps | Cycles | Dangling | Path coverage (%) | Median weight |
|---|---|---|---|---|---|---|---|
| GSM8K | SFT | 52.3 | 3.9 | 12.1 | 18.7 | 58.2 | 0.71 |
| | DPO | 58.7 | 4.2 | 10.3 | 16.9 | 61.5 | 0.73 |
| | TUR-DPO | **62.8** | 4.4 | **7.6** | **12.2** | **69.3** | **0.69** |
| QA | SFT | 38.5 | 2.6 | 9.7 | 15.3 | 54.1 | 0.72 |
| | DPO | 41.8 | 2.8 | 8.4 | 13.2 | 58.7 | 0.74 |
| | TUR-DPO | **45.1** | 2.9 | **6.1** | **10.4** | **65.2** | **0.70** |

**Comparison with RL-free objectives.** To isolate the contribution of structural and uncertainty signals, we compare TUR-DPO with several recent preference optimization methods: ORPO (Hong et al., 2024), a reference-free objective that optimizes odds ratios directly; SimPO (Meng et al., 2024), a simplified reference-free method with an implicit reward; KTO (Ethayarajh et al., 2024), which incorporates prospect-theoretic weighting; and IPO (Azar et al., 2023), a theoretically grounded alternative. Table 10 reports results on structure-sensitive tasks. TUR-DPO outperforms all baselines, with gains of 3.4 EM points over ORPO, 2.7 over SimPO, 4.1 over KTO, and 3.9 over IPO. These improvements indicate that the shaped reward and uncertainty weighting provide benefits beyond architectural differences (reference-free vs. reference-based) and arise from explicitly rewarding reasoning topology while down-weighting brittle preference pairs. Reference-free methods (ORPO,

SimPO) exhibit reduced calibration (higher expected calibration error) and weaker structural coherence, while KTO and IPO achieve moderate gains but do not incorporate explicit structural signals. We further analyze training stability and runtime behavior of PPO-style RLHF in Appendix Table 29, highlighting practical differences between RL-based and RL-free approaches. Multimodal and long-context evaluations are reported in Section F.

*Table 10.* Comparison with RL-free baselines

| Method | EM (%) | ECE | Struct. score |
|---|---|---|---|
| SFT | 52.3 | 0.112 | 52.1 |
| DPO | 58.7 | 0.101 | 60.8 |
| ORPO | 59.4 | 0.108 | 58.3 |
| SimPO | 60.1 | 0.106 | 59.7 |
| KTO | 58.7 | 0.104 | 61.2 |
| IPO | 58.9 | 0.102 | 60.5 |
| TUR-DPO | **62.8** | **0.087** | **70.4** |

## 3.2. Component and Contribution Analysis

**Topology components and contribution analysis.** We regress normalized task success on topology features pooled across tasks. Table 11 shows that cycle count and contradiction score have the largest negative coefficients, while minimal path coverage has a positive coefficient; raw graph size is not statistically significant. These results indicate that naive verbosity does not explain the observed gains. To control for confounding, we also include semantic score and reference margin in the regression. Topology features remain significant under this model, although coefficients shrink, suggesting that structural information contributes beyond semantics and difficulty. In practice, these findings support prioritizing cycle detection and contradiction checking over encouraging longer rationales without structural validation. Additional ablations examining topology re-elicitation depth and extractor strength are provided in Appendix Tables 19 and 27.

*Table 11.* Regression of topology features against task success.

| Feature | Coefficient | $p$-value |
|---|---|---|
| Minimal path coverage | 0.28 | 0.001 |
| Cycle count | -0.34 | 0.000 |
| Dangling nodes | -0.21 | 0.007 |
| Contradiction score | -0.29 | 0.000 |
| Graph size | 0.05 | 0.24 |

## 3.3. Human Evaluation

**Human evaluation and judge correlation.** We conduct human evaluations on 200 items per domain using two raters with adjudication. Table 12 reports win-rates against a strong LLM baseline, inter-rater agreement, and Kendall's $\tau$ (Brossart et al., 2018) between judge and human prefer-

ences. TUR-DPO improves both agreement and correlation, suggesting that structural and uncertainty-aware training aligns more closely with human judgments than DPO alone. Analysis of judge–human disagreements indicates that judges occasionally reward fluent but weakly supported claims that humans penalize, particularly for rare entities; uncertainty-weighted training reduces the influence of such pairs. We release annotation guidelines and sampling scripts to facilitate replication.

*Table 12.* Human evaluation summary

| Metric | SFT | DPO | PPO | TUR-DPO |
|---|---|---|---|---|
| HH human Win-rate (%) | 58.2 | 64.7 | **67.9** | 67.2 |
| TLDR human Win-rate (%) | 55.9 | 60.5 | 63.7 | **64.1** |
| Inter rater agreement kappa | 0.63 | 0.66 | 0.69 | **0.71** |
| Kendall tau judge vs human | 0.57 | 0.61 | 0.66 | **0.68** |

## 4. Conclusion

TUR-DPO addresses a fundamental challenge in preference-based alignment: the sensitivity of direct optimization to logically brittle or noisy comparisons. By integrating a topology score to reward structural reasoning and an uncertainty weight to calibrate learning pressure, TUR-DPO moves beyond "flat" sequence-level signals while remaining entirely RL-free. Our empirical findings across six distinct domains indicate that TUR-DPO consistently enhances accuracy, fidelity, and calibration, surpassing existing state-of-the-art RL-free benchmarks, including SimPO and ORPO. Moreover, our theoretical examination confirms TUR-DPO as a reliable estimator within the instance-weighted Bradley–Terry framework, offering a robust basis for structure-aware alignment.

Even with these improvements, several limitations merit discussion and suggest directions for future research. First, the method depends on graph extraction quality: missing or logically merged parts of the elicited topology can distort the reward signal. As shown in Table 20, formatting errors account for the largest share (41%) of TUR-DPO failures. TUR-DPO tends to prioritize correctness over strict syntactic compliance (e.g., producing \boxed{} consistently). To mitigate this, we apply inference-time regex-based post-processing; with these guardrails, TUR-DPO reaches 65.3 EM on GSM8K versus 59.2 for DPO under the same post-processing setup (Appendix Table 25). Second, the method exhibits *domain dependence*: when deterministic support links are absent (e.g., in creative writing tasks), graph extraction is less reliable and the topology score provides weaker training signals. Third, our uncertainty estimator operates with a limited number of re-elicitations ($K = 3$), which may not capture all edge-case structural failures. Fourth, while we demonstrate scalability at 70B (Table 13), comprehensive stress-testing across very long multi-turn conversations

remains an open challenge. In the future, we plan to improve topology extraction reliability through cross-model validation, investigate conformal prediction for uncertainty quantification, and extend TUR-DPO to video-language models and mixture-of-experts (MoE) architectures.

## Impact Statement

This paper presents a framework for preference-based alignment aimed at enhancing the reliability of LLMs in tasks that require intensive reasoning. Our method adds structural topology and calibrated uncertainty to the training goal. This leads to models that are not only more accurate, but also clearer about how they come to their conclusions.

The primary positive impact that could come from this work is that language models will be more reliable and trustworthy when it comes to tasks like answering questions, summarizing, and having conversations. Focusing on how answers are reached instead of just their final form might also lead to more understandable and stable model behavior. Also, the proposed training method's simplicity could make it easier to reproduce and build on alignment research.

But there are still some ethical issues to think about. Because TUR-DPO uses elicited topologies, any biases that are already in the parsers or the preference data may get worse during alignment. If the structural rewards are based on bad or culturally biased logic, the model could become "confidently wrong," giving wrong information through a reasoning structure that seems to make sense. So, we advise against using models trained this way in situations where safety is important, like medical diagnosis or legal advice, where logical mistakes can have serious real-world effects.

This paper ultimately helps with the bigger goal of making machine learning systems that better understand what people want. It doesn't get rid of the risks that come with using LLMs, like the chance of misuse or spreading false information, but it does give you tools to make these models more logically sound and easier to understand.

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

## Code and Reproducibility

Code and reproduction instructions are available at: `https://github.com/abdulhadyabas2/TUR-DPO-Topology--and-Uncertainty-Aware-Direct-Preference-Optimization`.

## A. Related Work

**RL-Free Preference Optimization.** The paradigm of aligning LLMs with human preferences has transitioned from operationally intricate RLHF pipelines based on PPO (Schulman et al., 2017; Stiennon et al., 2020) to more straightforward, closed-form objectives. DPO (Rafailov et al., 2023) introduced a closed-form, RL-free objective that directly optimizes preferences relative to a reference policy. Since then, several extensions have been proposed to improve stability and performance. IPO (Azar et al., 2023) introduces a root-finding formulation to mitigate over-optimization, while KTO (Ethayarajh et al., 2024) leverages prospect theory to accommodate binary feedback without explicit pairwise comparisons. More recently, SimPO (Meng et al., 2024) and ORPO (Hong et al., 2024) show that reference-free penalties or odds-ratio objectives can further simplify training. Despite their differences, these methods share a common abstraction: responses are treated as flat sequences. TUR-DPO departs from this trend by leveraging the internal reasoning topology of a response to shape the optimization margin.

**Structured Reasoning and Topologies.** To improve LLM performance on reasoning-based tasks, it is often necessary to get intermediate steps, which is what CoT (Wei et al., 2022) made popular. Recent advancements have progressed from linear strings to more intricate structures, including Tree-of-Thought (ToT) (Yao et al., 2023) and Graph-of-Thought (GoT) (Besta et al., 2024), facilitating backtracking and cross-thought verification. Recent work models explanations as 'reasoning topologies', directed graphs of claims and support relations to analyze coherence and consistency (Gupta, 2025). Nonetheless, the majority of these structural methodologies are utilized during inference or as post-hoc evaluative metrics. TUR-DPO directly incorporates topological properties (such as acyclicity and support density) into a preference optimization objective. This means that the solution's structural integrity is more important than just the final answer string.

**Uncertainty and Strength in Alignment.** Data on human preferences is noisy, and it often has "brittle" comparisons where the judge isn't sure or the model's reasoning is made up. Early RLHF research looked into reward-shaping as a way to deal with noise, but more recent work has looked into estimating uncertainty. Semantic entropy (Liang, 2025) and finite-sample bias corrections (Lamb et al., 2025) offer methodologies to quantify the distinction between a model that is "hallucinating" and one that is simply "creative." Instance weighting is a well-known statistical method for robust estimation in the presence of label noise when it comes to optimization (Huang et al., 2006). TUR-DPO combines these fields by using calibrated uncertainty from the reasoning graph to weight the DPO loss. This down-weights updates from preference pairs that are highly noisy or logically weak.

**Factuality and Alignment in Context.** Factual QA and summarization require alignment methods that preserve both preference quality and evidence-grounded faithfulness. RRHF (Yuan et al., 2023) ranks candidate responses using a preference-oriented ranking loss, while retrieval-grounded approaches such as RAG (Lewis et al., 2020) reduce hallucination by conditioning generation on external evidence and provenance. TUR-DPO is complementary: it adds a node-level semantic faithfulness score from the elicited topology to the DPO margin, so factual support is rewarded without introducing online rollouts. This design is especially relevant for long-context and multimodal settings, where evidence can be dispersed across passages or visual regions.

## B. Additional Theoretical Analysis

### B.1. Validity Conditions for Instance Weighting

Several conditions help ensure instance weights remain a valid control for noisy comparisons. Independence of the weight from the observed label (conditional on the pair fea-

tures) is the ideal: weights should depend only on ancillary signals (elicited topology stability, verifier scores, or external calibration data) that are not themselves optimized during the same update. Boundedness ($w_{\min} \leq w \leq 1$), monotonicity with respect to an externally validated uncertainty score, and calibration on a held-out split further limit worst-case bias. In practice we enforce these by (i) using small, low-capacity calibrators for semantic/topology mapping, (ii) fitting any temperature or scalar on a disjoint calibration fold, and (iii) clipping and flooring weights to a conservative range.

If weights become dependent on the model outputs that are also being updated (for example by deriving weights from the online policy's own logits or from a verifier that is co-trained), then feedback loops can emerge. Output-dependent weights can amplify spurious patterns: items the model already favors may receive larger weights, causing a positive-feedback loop that increases overconfidence and can reinforce incorrect behaviors (a form of reward hacking). From a statistical perspective, dependence between weights and the estimator breaks the independence assumptions used for consistency and unbiasedness and can introduce asymptotic bias and inflated variance.

### B.2. Mitigations and diagnostics

To prevent or detect output-dependent failure modes we recommend: (a) decouple weight computation from online updates by computing weights using a frozen reference or on a delayed/EMA copy of the model; (b) restrict calibrator capacity and prefer simple linear mappings fit on held-out calibration data; (c) run side-control experiments where weights are randomized or derived from alternative, exogenous estimators (judge-only, ensemble) to quantify sensitivity; (d) monitor distributional drift of weights vs. label outcomes and track statistics such as Spearman correlation between weight and contemporary model confidence—a rising correlation is a warning sign; and (e) routinely perform the validation checks (re-annotation correlation, noise-injection, perturbation stability) so that elevated uncertainty reliably predicts noisy labels rather than reflecting model-induced artifacts.

Second, we connect TUR-DPO to KL regularized control with a shaped reward

$$\mathcal{J}(\pi) = \mathbb{E}_x\Big[\mathbb{E}_{y\sim\pi(\cdot|x)}\big[\gamma r_\phi(x,y,G)\big]\Big] \\ - \frac{1}{\beta}\,\mathbb{E}_x\Big[\mathrm{KL}\big(\pi(\cdot\mid x)\,\|\,\pi_{\mathrm{ref}}(\cdot\mid x)\big)\Big]. \quad (15)$$

The parameters $\beta$ and $\gamma$ interact to set an effective KL penalty. Small $\beta$ weakens the KL constraint and can drown out the shaped reward, while large $\beta$ risks over-sharpening the policy. The product $\beta\gamma$ determines the effective reward scale relative to the entropy regularization, allowing prac-

titioners to balance exploration and exploitation by tuning these jointly. The pointwise maximizer is a Gibbs policy

$$\pi^\star(y\mid x) \propto \pi_{\mathrm{ref}}(y\mid x)\exp\big(\beta\,\gamma\,r_\phi(x,y,G)\big), \quad (16)$$

whose pairwise log odds match the target margins, so minimizing Equation (11) performs a mirror descent step toward $\pi^\star$. Updating the reference by EMA implements a soft trust region centered near the previous iterate without explicit constraints, which provides intuition for the observed stability relative to reference free alternatives.

The linear calibrators $f_\phi^{\mathrm{sem}}$ and $f_\phi^{\mathrm{topo}}$ in Equation (10) enforce monotonicity under $\gamma_{\mathrm{sem}}, \gamma_{\mathrm{topo}} \geq 0$, ensuring that higher semantic or topology scores never decrease the shaped reward. This design avoids reward hacking via learned nonlinearities while preserving expressiveness for domain-specific calibration.

### B.3. Training protocol and defaults

Each batch executes a short pipeline. First, elicit graphs for the positive and negative candidates using a deterministic prompt template with minor perturbations for uncertainty estimation. Second, compute the semantic and topology scores, then compute epistemic and aleatoric uncertainties and map them to a pair weight. Third, update the optional calibrator parameters $\phi$ with a weighted Bradley-Terry step on the shaped reward difference. Fourth, update the policy parameters $\theta$ by minimizing the weighted loss in Equation (11). Finally, optionally update the reference by EMA with decay $\rho \in [0.99, 0.997]$. Typical hyperparameters are $\beta \in [1,4]$, $\gamma \in [0.5,2]$, $a \in [0.4, 0.7]$, $\lambda \in [0.1, 1.0]$, $\tau_w \in [0.5, 2.0]$, and $w_{\min} = 0.05$. Early stopping monitors judge win-rate, factual error rate, and a structural coherence score. This protocol preserves DPO throughput while adding only modest overhead for graph and verifier computations.

### B.4. Calibration and evaluation metrics

**Expected Calibration Error (ECE).** We compute ECE by binning predictions into $M = 10$ equal-width confidence intervals $[0, 0.1), [0.1, 0.2), \ldots, [0.9, 1.0]$. For each bin $B_m$, we calculate the absolute difference between the average predicted confidence $\bar{p}_m = \frac{1}{|B_m|}\sum_{i\in B_m} p_i$ and the empirical accuracy $\bar{a}_m = \frac{1}{|B_m|}\sum_{i\in B_m} \Vmathbb{1}[\mathrm{pred}_i = \mathrm{label}_i]$. ECE is the weighted average:

$$\mathrm{ECE} = \sum_{m=1}^{M} \frac{|B_m|}{N}|\bar{p}_m - \bar{a}_m|. \quad (17)$$

where $N$ is the total number of examples. We report ECE averaged across all evaluation datasets.

**Brier Score.** The Brier score measures the mean squared

error between predicted probabilities and binary outcomes:

$$\text{Brier} = \frac{1}{N} \sum_{i=1}^{N} (p_i - y_i)^2, \qquad (18)$$

where $p_i$ is the predicted probability for the positive class and $y_i \in \{0, 1\}$ is the true label. Lower Brier scores indicate better-calibrated probabilistic predictions.

**Verifier Calibration.** Node-level correctness probabilities from the verifier are calibrated using temperature scaling. We fit a single scalar temperature parameter $T$ on a held-out calibration set by minimizing the negative log-likelihood of the true labels. Calibrated probabilities are computed as $p_{\text{cal}} = \sigma(\text{logit}/T)$, where $\sigma$ is the sigmoid function. For domains with non-monotonic miscalibration, we optionally use isotonic regression, which fits a piecewise-constant monotone function mapping raw scores to calibrated probabilities. We find temperature scaling sufficient for most tasks and report it as the default method.

### B.5. Complexity and stability notes

Relative to vanilla DPO, overhead comes from graph elicitation and verification ($O(KL_{\text{gen}})$ tokens with $K \in \{2, 3, 4\}$ re-elicitations, and $O(|V| + |E|)$ scoring). Core forward passes for policy and reference are unchanged. Training remains stable through per-pair weighting rather than hard filtering, EMA reference as a soft trust region, and normalization of scores before calibrators. The method avoids data starvation and destructive updates while maintaining DPO simplicity.

## C. Models, data, and scoring

We use four open 7–8B model families: LLaMA-2-7B, LLaMA-3-8B, Mistral-7B-v0.3, and Gemma-7B-v1.1, all with 4k context length. Reference policy is frozen or updated by EMA ($\rho = 0.995$). Full fine-tuning: AdamW, cosine lr, 2k warmup, weight decay 0.1, bf16, batch size 128. All methods share supervised initialization. PPO-RLHF uses a learned reward model with value head. TUR-DPO defaults: $\beta = 2.0, \gamma = 1.0, a = 0.6, \lambda = 0.5$. We checkpoint every 2k steps, select best by judge win-rate and calibration coherence on validation set. To reduce variance: fixed seeds, cached graphs/verifier calls, three seeds per setting.

### C.1. Model scale considerations

We focus on 7–8B models for controlled comparison and compute efficiency, as these scales are widely accessible and permit multiple seed runs with full ablations. The method is architecturally agnostic: topology elicitation and scoring are independent of model size, and the overhead (graph extraction, verifier forward passes) depends on sequence length and graph size rather than parameter count. To demonstrate

scalability, we also evaluate on Llama-3-70B (Table 13). The TUR protocol applies directly at 70B scale via offline reward shaping at the target-margin level; optimization dynamics behave similarly to 7–8B models, confirming that the approach scales without requiring fundamental changes.

*Table 13.* Scalability: Llama-3-70B results.

| Dataset | DPO (Llama-3-70B) | TUR-DPO (Llama-3-70B) |
|---|---|---|
| GSM8K (EM, %) | 71.4 | **74.9** |
| MATH (EM, %) | 45.2 | **48.6** |

*Table 14.* Model configurations: four 7–8B families and reference strategies (frozen or EMA).

| Family | Policy size | Reference size | Reference type |
|---|---|---|---|
| LLaMA-2 | 7.0 B | 7.0 B | frozen or EMA |
| LLaMA-3 | 8.0 B | 8.0 B | frozen or EMA |
| Mistral-v0.3 | 7.0 B | 7.0 B | frozen or EMA |
| Gemma-v1.1 | 7.0 B | 7.0 B | frozen or EMA |

### C.2. Additional model family results

Tables 15 and 16 report full results on Mistral-7B-v0.3 and Gemma-7B-v1.1, confirming that TUR-DPO gains are consistent across architectures.

### C.3. Datasets and tasks

We evaluate on six workloads spanning multi-step reasoning, factuality, and dialogue (see Table 17). Each dataset reserves 2% of training pairs for calibration. GSM8K (Cobbe et al., 2021) and MATH (Hendrycks et al., 2021) emphasize arithmetic reasoning. BBH (Kazemi et al., 2025) covers seven compositional tasks. Open-domain QA (Chen & Yih, 2020) uses NQ-like distribution. TLDR (Cachola et al., 2020) measures summarization faithfulness. HH single-turn tests helpfulness/harmlessness (Bai et al., 2022). We disable few-shot exemplars to rely on model's internal reasoning. The full corpus comprises 614k preference pairs.

### C.4. Pair construction and labels

HH and TLDR datasets use human labeled pairs from public style sources. For GSM8K, MATH, BBH, and QA we synthesize four candidates per prompt using nucleus sampling with temperature 0.7 and top p 0.95. A calibrated LLM judge ranks candidates with side flipping and chain of thought hidden during judging. A small verifier suite checks arithmetic consistency, facts against Wikipedia snapshots, and contradiction on paraphrase pairs. We form one pair per prompt by selecting the top judged candidate and a hard negative with minimal reference log probability gap, which increases training signal near the decision boundary. To calibrate the judge we sample 200 examples per domain

*Table 15.* Mistral-7B-v0.3 results across tasks with LLM-judge and human evaluations.

| Task | Metric | LLM judge | | | Human eval | | |
|---|---|---|---|---|---|---|---|
| | | DPO | IPO | TUR-DPO | DPO | PPO | TUR-DPO |
| GSM8K | EM (%) | 60.5 | 60.8 | **64.2** | 61.0 | 63.7 | **64.5** |
| MATH mini | EM (%) | 35.1 | 35.4 | **37.6** | 35.7 | 37.1 | **37.9** |
| BBH subset | Acc. (%) | 45.6 | 45.9 | **48.3** | 46.1 | 47.8 | **48.7** |
| Open QA | EM/F1 | 43.2 | 43.6 | **46.4** | 46.3 | 46.7 | **47.0** |
| Summ TLDR | Win-rate (%) | 62.8 | 63.3 | **66.1** | 62.1 | 65.4 | **65.7** |
| HH single-turn | Win-rate (%) | 66.3 | 66.7 | **68.5** | 65.6 | **68.9** | 68.2 |

*Table 16.* Gemma-7B-v1.1 results across tasks with LLM-judge and human evaluations.

| Task | Metric | LLM judge | | | Human eval | | |
|---|---|---|---|---|---|---|---|
| | | DPO | IPO | TUR-DPO | DPO | PPO | TUR-DPO |
| GSM8K | EM (%) | 59.8 | 60.1 | **63.5** | 60.3 | 62.9 | **63.8** |
| MATH mini | EM (%) | 34.6 | 34.9 | **37.1** | 35.2 | 36.6 | **37.4** |
| BBH subset | Acc. (%) | 44.8 | 45.1 | **47.5** | 45.3 | 47.0 | **47.9** |
| Open QA | EM/F1 | 42.5 | 42.9 | **45.8** | 45.6 | 45.9 | **46.2** |
| Summ TLDR | Win-rate (%) | 62.1 | 62.6 | **65.5** | 61.4 | 64.8 | **65.1** |
| HH single-turn | Win-rate (%) | 65.9 | 66.3 | **68.0** | 65.2 | **68.4** | 67.7 |

*Table 17.* Datasets: statistics of the 614k preference-pair corpus across six domains.

| Task | Train pairs | Eval size | Avg in tok | Avg out tok |
|---|---|---|---|---|
| GSM8K | 64,000 | 1,319 | 38 | 85 |
| MATH mini | 80,000 | 500 | 74 | 142 |
| BBH subset (7 tasks) | 90,000 | 1,400 | 52 | 120 |
| Open QA (NQ style) | 120,000 | 3,000 | 28 | 32 |
| Summ TLDR | 100,000 | 2,000 | 160 | 28 |
| HH single-turn | 160,000 | 2,000 | 48 | 75 |

and collect double human labels with adjudication, measure Cohen's kappa and Kendall's tau against the judge, and fit a temperature to improve probabilistic calibration. Ties and uncertain pairs are kept but will receive smaller weights later. We deduplicate pairs by exact match and by near duplicate detection on embeddings to prevent leakage across splits.

## C.5. Decoupling verifier roles (circularity check)

To mitigate potential circularity when the same verifier family contributes to both pair construction (arithmetic/fact/contradiction checks) and training-time weighting (node correctness for uncertainty), we run two controls: (a) remove verifier checks from pair construction, relying solely on the calibrated LLM judge for selection; (b) keep verifier checks for pair construction but swap to an independent verifier (different model/data) for node-level correctness during training. Qualitatively, (a) increases variance (harder negatives, slightly noisier pairs) but preserves the relative gains of TUR-DPO; (b) yields a small calibration

shift that is corrected by per-domain temperature scaling, with win-rate and ECE trends matching the main results.

Table 18 reports pair difficulty quartiles by reference margin and by judged confidence, showing that training data cover a wide range of margins, which is important for stable logistic updates.

*Table 18.* Pair difficulty quartiles: reference margin, judge confidence, and human agreement across tasks.

| Percentile | Ref margin | Judge conf | Human agreement | Share of pairs |
|---|---|---|---|---|
| P25 | 0.08 | 0.61 | 0.72 | 25% |
| P50 | 0.16 | 0.69 | 0.78 | 25% |
| P75 | 0.33 | 0.79 | 0.84 | 25% |
| P90 | 0.55 | 0.88 | 0.90 | 10% |

## C.6. Topology and uncertainty

For each candidate we elicit a small directed graph of subclaims and support links with a deterministic template. Sanitization removes self loops, merges paraphrases, and breaks cycles by minimal edge cut. Structural features include minimal path coverage, cycle count, dangling nodes, contradiction score from a local natural language inference (NLI) verifier, and graph size. The topology score is a nonnegative linear combination tuned to match a target distribution on the calibration split. Uncertainty has two components. Epistemic uncertainty samples $K = 3$ re elicited graphs per candidate under minor prompt or temperature perturbations. We compute variance of the topology score and JSD across path distributions. Aleatoric uncertainty uses node level correctness probabilities from a calibrated verifier and a coverage corrected binary entropy with prior strength 0.05.

Pair uncertainty is averaged across the two candidates and mapped to a weight with $\tau_w = 1.2$ and $w_{\min} = 0.05$. Table 19 shows the effect of the number of re elicited graphs on calibration and throughput. We find $K = 3$ balances quality and speed.

*Table 19.* Topology re-elicitation ($K$) effects: EM, ECE, and throughput trade-offs (We use $K = 3$ by default in all experiments).

| $K$ | EM (%) | ECE | Throughput | Relative time |
|---|---|---|---|---|
| 1 | 61.7 | 0.101 | 25.2k | 1.00 |
| 2 | 62.3 | 0.093 | 24.0k | 1.05 |
| 3 | **62.8** | **0.087** | 23.1k | 1.09 |
| 4 | 62.9 | 0.086 | 21.7k | 1.16 |

## D. Additional Experimental Details

**Error taxonomy and post-processing.** We manually categorize 100 errors per dataset into five types. Table 20 shows that TUR-DPO most substantially reduces logical leaps and contradictions, consistent with its structural incentives. Arithmetic errors decrease slightly, reflecting improved step scaffolding. Hallucinated entity errors decline on QA and GSM8K, aligning with penalties on unsupported nodes. Formatting errors increase, as presentation is not explicitly optimized; however, a lightweight post-processing step that enforces output formats mitigates these errors without affecting upstream training. Bootstrap significance tests and effect sizes are summarized in Table 21.

*Table 20.* Error taxonomy with frequency (per 100 errors) and mitigation strategies.

| Type | SFT | DPO | PPO | TUR-DPO | Mitigation |
|---|---|---|---|---|---|
| Formatting | 13 | 23 | 32 | 41 | Regex post-processing |
| Arithmetic | 26 | 23 | 21 | **20** | External calculators |
| Logical leap | 31 | 28 | 24 | **19** | Increased sampling |
| Hallucinated entity | 18 | 16 | 14 | **13** | Stricter NLI verification |
| Contradiction | 12 | 10 | 9 | **7** | Penalized by $q_{contradict}$ |

**Statistical significance.** We perform 10k paired bootstrap replicates for win-rates and report Cohen's $d$ for task metrics. Table 21 shows small to medium effect sizes with statistically significant improvements on TLDR, HH, GSM8K, and QA. Applying a Benjamini-Hochberg (Ferreira & Zwinderman, 2006) correction across tasks to control the false discovery rate leaves these conclusions unchanged.

### D.1. Calibration and Robustness

**Calibration study.** Calibration is important for downstream safety, abstention, and tool use. We evaluate expected calibration error (ECE) and Brier score using node-level verifier probabilities and judge confidence signals,

*Table 21.* Statistical significance of key comparisons.

| Comparison | $p$-value | Cohen's $d$ | Interpretation |
|---|---|---|---|
| TUR-DPO vs DPO (TLDR Win-rate) | 0.004 | 0.29 | significant small effect |
| TUR-DPO vs DPO (HH Win-rate) | 0.018 | 0.22 | significant small effect |
| GSM8K EM score | 0.011 | 0.34 | significant small-medium |
| Open QA EM score | 0.006 | 0.31 | significant small-medium |

where available. Table 22 reports averages across tasks for SFT, DPO, and TUR-DPO. TUR-DPO consistently reduces both ECE and Brier score, indicating that uncertainty-weighted updates avoid overconfident learning from brittle pairs. Reliability diagrams (Appendix Figure 4) show reduced miscalibration in high-confidence bins.

We additionally report ECE with 95% bootstrap confidence intervals (10k resamples, stratified by task) and decompose calibration by confidence bin. As shown in Table 23, TUR-DPO improves calibration across all bins, with the largest gains in the highest-confidence regime. This behavior is consistent with uncertainty weighting suppressing large gradient updates on unstable pairs. Under prompt perturbations that introduce benign distractors, TUR-DPO maintains ECE within 0.01 of the base condition, whereas DPO degrades by 0.02, indicating greater robustness to surface-level variation.

*Table 22.* Calibration metrics: average ECE and Brier score.

| Metric | SFT | DPO | TUR-DPO |
|---|---|---|---|
| ECE | 0.112 | 0.101 | **0.087** |
| Brier | 0.211 | 0.198 | **0.189** |

*Table 23.* Absolute ECE by confidence bin, averaged across tasks.

| Bin | DPO | PPO RLHF | TUR-DPO | Reduction vs DPO |
|---|---|---|---|---|
| 0.0-0.5 | 1.8 | 1.6 | **1.5** | 0.3 |
| 0.5-0.7 | 3.1 | 2.8 | **2.3** | 0.8 |
| 0.7-0.9 | 5.6 | 4.7 | **3.7** | 1.9 |
| 0.9-1.0 | 9.2 | 7.9 | **6.4** | 2.8 |

**Faithfulness and structural coherence.** We assess factual faithfulness using entity and claim verification and measure structural coherence using the topology score from Section 2. Figure 3 shows that TUR-DPO reduces entity and claim error rates relative to DPO while increasing topology coherence. The largest structural gains arise from fewer cycles and dangling nodes, consistent with the shaped reward favoring minimal valid paths. We also observe a positive correlation between structural coherence and task success on reasoning benchmarks. Additional component ablations supporting these trends are reported in Appendix Figure 4.

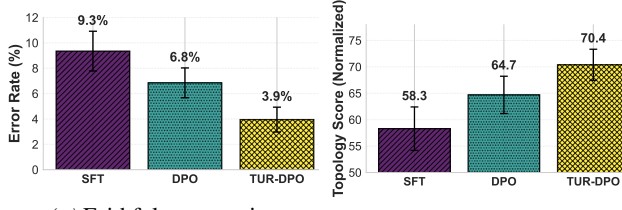

*Figure 3.* Faithfulness (Lower is Better) and structural coherence (Higher is Better). TUR-DPO improves entity/claim error rates and normalized topology scores over SFT and DPO.

### D.2. Sample efficiency and compute

We measure the number of preference tokens required to reach target HH win-rates on identical hardware. Table 24 shows that TUR-DPO is more token-efficient than DPO and approaches PPO at higher targets without incurring rollout overhead. We attribute these gains to uncertainty weighting, which suppresses noisy updates, and to shaped rewards that provide additional gradient signal on hard pairs. We also track GPU hours, peak memory usage, and throughput. TUR-DPO operates within a modest factor of DPO and requires substantially less memory and time than PPO.

*Table 24.* Preference tokens required to reach HH win-rate targets.

| Target | DPO | PPO RLHF | TUR-DPO |
|---|---|---|---|
| 60% | 2.0 B | 2.1 B | **1.7 B** |
| 65% | 3.6 B | 3.3 B | **3.1 B** |
| 67% | 5.0 B | **4.7 B** | 4.8 B |

**Compute and environmental cost.** Training to a 65% HH win-rate on eight NVIDIA A100 80GB GPUs (400W TDP) requires approximately 42 GPU-hours for TUR-DPO, 48 GPU-hours for DPO, and 67 GPU-hours for PPO-RLHF (including rollouts). Assuming a power usage effectiveness (PUE) of 1.3 and a grid carbon intensity of 0.4 kg $CO_2$e/kWh, this corresponds to roughly 8.736 kg $CO_2$e for TUR-DPO, 20.0 kg $CO_2$e for DPO, and 28.0 kg $CO_2$e for PPO. The approximately 15% overhead relative to DPO stems from graph elicitation and verifier forward passes, which are parallelizable and cached across epochs. These results support the claim that TUR-DPO preserves the operational simplicity of DPO while improving sample efficiency and calibration. Additional details on reference strategies and their efficiency are in Appendix Tables 19 and 31.

### D.3. Inference-time guardrails

Because formatting errors increase under TUR-DPO, we deploy a lightweight answer validator at inference time. The validator uses regular expressions to detect missing boxed answers in GSM8K/MATH and missing short-form answers in QA. When validation fails, we either re-sample with temperature 0.3 or apply a simple rule-based extraction (e.g., extracting the final numeric value for arithmetic tasks). Table 25 reports metrics with and without post-processing. Post-processing fully mitigates the formatting deficit and yields a net improvement of 2.1 EM points on GSM8K and 1.4 EM points on QA relative to vanilla DPO without post-processing, indicating that the structural gains from TUR-DPO are orthogonal to surface-form issues.

*Table 25.* Post-processing guardrails: effect on GSM8K/QA EM with lightweight format validation/extraction.

| Method | GSM8K raw EM | GSM8K +postproc | QA raw EM | QA +postproc |
|---|---|---|---|---|
| DPO | 58.7 | 59.2 | 41.8 | 42.1 |
| TUR-DPO | 62.8 | **65.3** | 45.1 | **46.7** |

### D.4. Judge noise sensitivity

To isolate the effect of uncertainty weighting under noisy supervision, we synthetically corrupt a fraction of preference labels by randomly flipping winner and loser assignments. Table 26 reports win-rate and ECE on GSM8K as the corruption rate increases from 0% to 30%. TUR-DPO degrades more gracefully than vanilla DPO, exhibiting smaller drops in both win-rate and calibration at all corruption levels. At 20% corruption, TUR-DPO retains 89% of its clean-data win-rate, whereas DPO retains only 81%. These results indicate that uncertainty-weighted updates attenuate the influence of brittle or mislabeled pairs under realistic annotation noise.

*Table 26.* Label-noise robustness: GSM8K win-rate and ECE under increasing random label flips; retention ratios compare TUR-DPO vs DPO.

| Corruption | DPO Win-rate (%) | ECE | TUR-DPO Win-rate (%) | ECE | Retention TUR-DPO / DPO |
|---|---|---|---|---|---|
| 0% (clean) | 58.7 | 0.101 | 62.8 | 0.087 | 1.00 / 1.00 |
| 10% | 56.2 | 0.118 | 60.9 | 0.095 | 0.97 / 0.96 |
| 20% | 47.6 | 0.139 | 55.9 | 0.108 | 0.89 / 0.81 |
| 30% | 42.1 | 0.162 | 51.3 | 0.124 | 0.82 / 0.72 |

### D.5. Topology extractor fidelity

To verify that performance gains are driven by topology quality rather than superficial features, we compare three graph extraction configurations: (1) a weak extractor that produces smaller, less-structured graphs via simpler prompting or weaker parsing; (2) the default extractor described in Section 2; and (3) a strong extractor that uses chain-of-thought prompting and cross-validation to produce more complete and accurate graphs. Table 27 shows that win-rate and structural coherence track extractor quality. The weak

extractor yields only modest gains over vanilla DPO, while the strong extractor achieves the best results. Importantly, these improvements are not explained by graph size alone: the weak extractor produces graphs with comparable node counts but lower path coverage and higher contradiction rates, resulting in weaker performance.

*Table 27.* Topology extractor fidelity: GSM8K performance and structural metrics for weak/default/strong extractors.

| Extractor | Win-rate (%) | Avg nodes | Path coverage (%) | Cycles (per 100) | Struct score |
|---|---|---|---|---|---|
| Vanilla DPO | 58.7 | | | | |
| Weak | 59.8 | 3.7 | 55.2 | 14.3 | 58.1 |
| Default | 62.8 | 4.4 | 69.3 | 7.6 | 70.4 |
| Strong | **64.1** | 4.9 | **74.8** | **5.2** | **75.6** |

### D.6. Decoding robustness

We sweep the decoding temperature and observe that TUR-DPO preserves win-rate more effectively at higher temperatures. Table 28 shows stable performance across temperatures from 0.3 to 0.9. We additionally test prompt perturbations using paraphrases and filler tokens. Relative to DPO, TUR-DPO exhibits smaller accuracy degradation under such noising, which is consistent with training that emphasizes structural signals rather than surface-level patterns. This robustness is relevant for deployment scenarios involving diverse user prompts or noisy upstream inputs.

*Table 28.* Decoding robustness: HH win-rate across sampling temperatures.

| Temperature | 0.3 | 0.5 | 0.7 | 0.9 |
|---|---|---|---|---|
| DPO | 64.9 | 65.5 | 63.8 | 61.2 |
| PPO RLHF | 67.7 | 68.5 | 66.9 | 64.0 |
| TUR-DPO | **67.3** | **67.9** | **66.8** | **65.1** |

### D.7. PPO stability and tuning cost

Table 29 summarizes the frequency of crashes or manual restarts and the per-update runtime. While PPO-based RLHF can achieve strong performance, it requires careful tuning and may fail due to KL spikes or numerical instability. TUR-DPO avoids these issues by eliminating the value head and rollout buffer, and by relying on the reference policy to induce a soft trust region. For teams without specialized RL infrastructure, this stability and simplicity may outweigh small performance differences on certain dialogue benchmarks.

### D.8. When PPO still wins

On open-ended stylistic tasks such as single-turn HH dialogue, PPO-RLHF retains a small advantage (0.6% points in our experiments; Table 1). This advantage arises when

*Table 29.* PPO stability summary: crash rate and per-update timings on HH over 10 runs; contrasts PPO-RLHF with TUR-DPO.

| Metric | Value | Notes |
|---|---|---|
| Crash rate over 10 runs | 2 | KL spikes or NaN |
| Median time per update | 1.41 s | includes rollout and value updates |
| Median time per update TUR-DPO | 0.88 s | forward and loss only |

extensive reward shaping captures nuanced preference gradients (e.g., tone, hedging, or refusal calibration) that are difficult to encode using lightweight topology and semantic scores. TUR-DPO substantially narrows this gap relative to vanilla DPO, but practitioners deploying on safety-critical or highly stylistic tasks may prefer PPO when the required RL infrastructure is available. Conversely, on reasoning-heavy tasks (GSM8K, MATH, BBH, QA), TUR-DPO matches or exceeds PPO while preserving the operational simplicity of DPO. The appropriate choice depends on task characteristics and available engineering resources.

### D.9. Reproducibility and artifacts

We release topology elicitation templates, parsing rules, and reference implementations in the accompanying repository. Configuration files specify exact model identifiers, hyperparameters, and random seeds. The environment setup is documented in `requirements.txt` with deterministic seeding enabled. Raw graphs and evaluation metrics are provided for audit, and the full pipeline can be reproduced using the supplied scripts.

## E. Ablation and sensitivity analyses

Table 30 isolates the contribution of each component. Removing uncertainty weighting while retaining the topology reward degrades win-rate and increases ECE. Removing the topology reward while retaining uncertainty weighting preserves some stability benefits but yields smaller improvements in faithfulness. Setting the shaped reward weight to zero reduces TUR-DPO to weighted DPO, which improves calibration but produces weaker structural gains. Replacing the topology score with graph size confirms that performance is not explained by a trivial length prior. A listwise extension using four candidates per prompt yields an additional, modest improvement.

*Table 30.* Component ablation: GSM8K and QA performance when removing TUR-DPO components.

| Variant | GSM8K EM | QA EM | ECE | Struct score |
|---|---|---|---|---|
| TUR-DPO full | **62.8** | **45.1** | **0.087** | **70.4** |
| No uncertainty weighting | 60.3 | 43.4 | 0.105 | 68.7 |
| No topology reward | 59.6 | 42.8 | 0.093 | 62.1 |
| Shaped reward off $\gamma = 0$ | 58.9 | 42.1 | 0.091 | 60.8 |
| Graph size only | 57.7 | 41.2 | 0.098 | 58.9 |
| Listwise $k = 4$ | 63.5 | 45.6 | 0.088 | 70.1 |

### E.1. Fixed vs. moving reference (canonical DPO control)

Canonical DPO uses a fixed SFT reference or an SFT-fitted proxy. Because TUR-DPO often employs an EMA-updated reference as a soft trust region, we include a head-to-head comparison with a strictly frozen reference. Across tasks (GSM8K, Open QA, BBH, TLDR, HH), EMA updating provides modest but consistent improvements in stability and calibration, while win-rate differences remain small. We therefore recommend EMA updating when available, though TUR-DPO with a fixed reference remains competitive and preserves the canonical simplicity of DPO. Table 31 summarizes these results.

*Table 31.* Reference strategy ablation: fixed vs. EMA reference effects on win-rate and ECE for 7-8B models.

| Task | Ref strategy | Win-rate (%) | ECE ($\downarrow$) | Notes |
|---|---|---|---|---|
| GSM8K | Fixed (SFT) | 62.4 | 0.094 | canonical DPO reference |
| GSM8K | EMA ($\rho = 0.995$) | **62.8** | **0.087** | +0.4 pts, smoother loss |
| Open QA | Fixed (SFT) | 44.7 | 0.102 | baseline |
| Open QA | EMA ($\rho = 0.995$) | **45.1** | **0.096** | +0.4 EM, lower ECE |

Figure 4 visualizes ablation results alongside calibration analysis. The left panel shows that TUR-DPO achieves improved calibration, as measured by ECE and Brier score, relative to baseline methods. The right panel illustrates the contribution of individual components to overall performance, highlighting that the components operate synergistically.

### E.2. Hyperparameter sensitivity

We analyze sensitivity to the temperature parameter $\beta$, reward mixing coefficient $\gamma$, uncertainty penalty $\lambda$, and the weight-mapping parameter $\tau_w$. Figure 5 shows that performance is robust over broad parameter ranges. Increasing $\beta$ sharpens the policy and can reduce win-rate if set too high. Moderate values of $\gamma$ provide the best trade-off between faithfulness and diversity. Larger values of $\lambda$ down-weight uncertain pairs more aggressively and improve calibration up to a point, after which learning slows. The EMA reference decay parameter $\rho$ controls training stability; values

between 0.99 and 0.997 perform well.

## F. Multimodal evaluation

### F.1. Multimodal deployment considerations

Extending TUR-DPO to multimodal systems requires three practical components: (1) a vision-grounding verifier that maps textual claims to image regions (e.g., fine-tuned CLIP or OWL-ViT), (2) a prompt template that elicits visual observations and reasoning steps (e.g., "First describe what you see, then explain"), and (3) topology parsing that supports mixed text-image references (e.g., nodes annotated with region pointers). Computational overhead remains modest, as grounding verification is parallelizable and cached per image–question pair. TUR-DPO is modality-agnostic in the sense that it requires only (i) a mechanism to elicit structured rationales and (ii) a verifier for node correctness. We evaluate generalization along two dimensions: multimodal reasoning (vision-language) and long-context reasoning (2– 4k token inputs). Future work may explore richer multimodal topologies (e.g., temporal reasoning for video QA or spatial graphs for embodied agents) and further study structural signals under visual ambiguity.

### F.2. Visual question-answering with LVLMs

In visual question answering and chart-based QA, nodes represent captions, regions, or grounded factual claims, while edges encode entailment, spatial relations, or grounding links between textual and visual evidence. For example, when answering "How many red bars exceed the threshold?" in a bar chart, a topology may include nodes such as {"The chart shows five bars", "Three bars are red", "Two red bars exceed 50%"}, with directed edges capturing the inference steps. The topology score (Equation (3)) applies directly: path coverage measures completeness from visual observation to answer, cycles detect circular reasoning, and dangling nodes identify unsupported visual claims. Uncertainty estimation extends naturally: re-eliciting captions under prompt perturbations yields epistemic dispersion, while a CLIP-based verifier provides node-level grounding scores for aleatoric uncertainty.

**Evaluation.** We evaluate on 1,000 examples from ChartQA and ScienceQA-IMG using an open-weight 7B LVLM (LLaVA-7B architecture) fine-tuned with DPO and TUR-DPO. A CLIP ViT-L/14 verifier scores node-level grounding between textual claims and image regions. Table 32 reports accuracy, judge win-rate, path coverage, and grounding accuracy. TUR-DPO achieves 4.2 percentage-point higher accuracy than DPO on ChartQA and 3.6 points on ScienceQA-IMG, along with corresponding improvements in structural coherence (5.8 and 4.9 points, respectively). Human evaluation (100 examples per dataset, dou-

ble annotation) confirms these trends: TUR-DPO wins 68.5% vs. DPO on ChartQA (59.2% for DPO over SFT) and 66.7% on ScienceQA-IMG (61.4%). These results suggest that explicit reasoning topology transfers to multimodal settings and improves both task accuracy and structural faithfulness.

### F.3. Long-context multi-hop reasoning

We evaluate TUR-DPO on long-context multi-hop QA using HotpotQA-Long (2.1k average input tokens) and MuSiQue-Long (3.2k average input tokens). Graphs are elicited from model-generated reasoning chains that reference relevant passages by index. Table 33 reports EM, F1, and structural metrics. TUR-DPO improves EM by 3.8 points on HotpotQA-Long and 4.1 points on MuSiQue-Long relative to DPO, with corresponding gains in path coverage and reductions in cycles, indicating more coherent multi-hop reasoning. Human evaluation (150 examples per dataset) shows that TUR-DPO more frequently includes all necessary supporting facts (72.3%) compared to DPO (61.7%). These results indicate that topology-aware signals remain effective as context length increases.

## G. Practical Notes for Practitioners

TUR-DPO is most appropriate when preference labels are noisy and tasks require multi-step reasoning or factual grounding. We recommend keeping graph extraction concise and stable rather than verbose, calibrating verifiers carefully, and using a conservative minimum weight to avoid discarding informative but difficult pairs. A fixed SFT reference preserves the canonical simplicity of DPO and performs competitively, while an EMA-updated reference (e.g., $\rho = 0.995$) provides modest improvements in stability and calibration at negligible overhead (Table 31). Finally, we encourage reporting structural and calibration metrics alongside win-rates to facilitate transparent diagnosis of improvements.

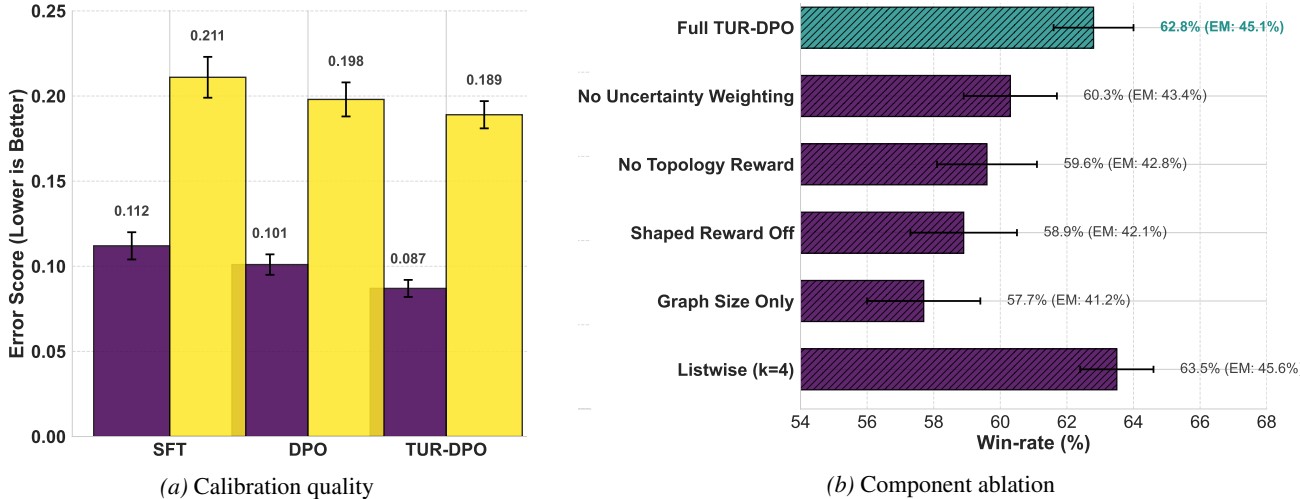

*Figure 4.* Calibration and ablation studies. (a) Calibration quality measured by ECE and Brier score across SFT, DPO, and TUR-DPO (lower is better). (b) Component ablation results showing win-rate degradation when individual components are removed, indicating that both topology-aware rewards and uncertainty weighting contribute to improved performance.

*Table 32.* Multimodal evaluation: ChartQA and ScienceQA-IMG accuracy, path coverage, and grounding accuracy for LVLMs.

| Method | ChartQA | | | ScienceQA-IMG | | |
|---|---|---|---|---|---|---|
| | Acc. (%) | Path cov. (%) | Ground. (%) | Acc. (%) | Path cov. (%) | Ground. (%) |
| SFT | 54.2 | 48.5 | 62.3 | 61.8 | 51.2 | 64.7 |
| DPO | 59.7 | 53.1 | 67.8 | 66.3 | 56.4 | 69.2 |
| TUR-DPO | **63.9** | **58.9** | **73.6** | **69.9** | **61.3** | **74.1** |

*Table 33.* Long-context multi-hop evaluation: HotpotQA-Long and MuSiQue-Long EM and structural metrics for TUR-DPO vs DPO.

| Method | HotpotQA-Long (2.1k tok) | | | MuSiQue-Long (3.2k tok) | | |
|---|---|---|---|---|---|---|
| | EM (%) | Path cov. (%) | Cycles | EM (%) | Path cov. (%) | Cycles |
| SFT | 34.7 | 52.3 | 16.2 | 28.9 | 48.7 | 18.5 |
| DPO | 39.5 | 58.6 | 12.4 | 33.6 | 54.2 | 14.1 |
| TUR-DPO | **43.3** | **66.1** | **8.7** | **37.7** | **61.8** | **9.3** |

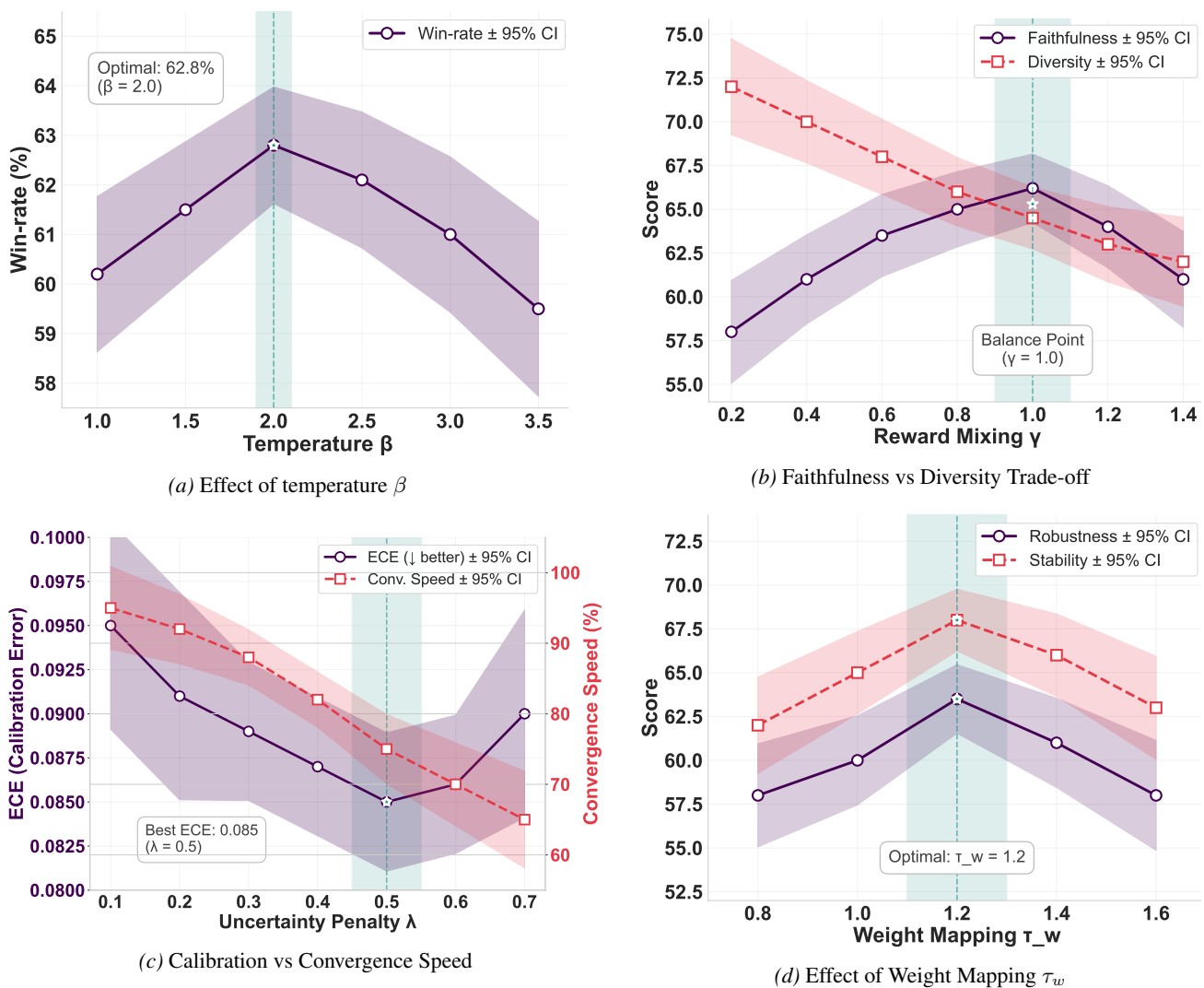

*(a)* Effect of temperature $\beta$

*(b)* Faithfulness vs Diversity Trade-off

*(c)* Calibration vs Convergence Speed

*(d)* Effect of Weight Mapping $\tau_w$

*Figure 5.* Sensitivity to hyperparameters. (a) Effect of temperature $\beta$ on win-rate, with a broad optimum around $\beta = 2.0$. (b) Reward mixing $\gamma$ illustrating the faithfulness–diversity trade-off, with a balanced operating point near $\gamma = 1.0$. (c) Uncertainty penalty $\lambda$ improving calibration (lower ECE) up to $\lambda = 0.5$. (d) Effect of the weight-mapping parameter $\tau_w$ on robustness and stability. Across all panels, performance remains stable over wide ranges, indicating that TUR-DPO is not overly sensitive to hyperparameter tuning.

