# OpenReview forum: "TUR-DPO: Topology- and Uncertainty-Aware Direct Preference Optimization"
_ICML.cc/2026/Conference — ICML 2026 regular_

### Official Review · Reviewer_9NhZ · 2026-03-09

**Soundness:** 3
**Presentation:** 2
**Significance:** 3
**Originality:** 3
**Overall Recommendation:** 4
**Confidence:** 3

**Summary:**

The current DPO can suffer from noisy preference pairs. So in this paper, they propose TUR-DPO, which factorizes semantic faithfulness, utility, and topology quality into a learnable reward and incorporate into an uncertainty-weighted DPO objective (by augmenting the DPO logits). Extensive experiments show its effectiveness.

**Compliance With Llm Reviewing Policy:**

Affirmed.

**Final Justification:**

Thanks for the response, which resolve my original concerns, I will maintain my current positive score.

**Key Questions For Authors:**

1. Is it a minor? On line 082 "semantically sound solutions (1)", what does (1) here refer to?
2. Am I omitting anything? What exactly are the model families you use for the open 7-8B models?
3. Given a response, how do you extract a graph and decompose the text into atomic statements and link supports?

**Limitations:**

yes

**Strengths And Weaknesses:**

**Strengths:**
1. It is an interesting idea to add reasoning structures into the DPO framework, and framing it with topology is clever, so that we can consider the quality within the reasoning process rather than only the final answer's quality.
2. They provide a theoretical analysis to rationalize their designed method and enable a deeper understanding.
3. Experiments are extensive, using different model families and on various benchmarks.


**Weaknesses:**
More details are needed, at least for me to clearly understand what is going on for the method design details and experimental setup. See questions.

---

> ### Author Rebuttal · Authors · 2026-03-29
>
> ## Response to Reviewer 1. We appreciate your review.
>
> ### A. Graph extraction and pipeline design.
>
> **Reviewer:** *Weak: Detailed Method design: Rev1 Q3 & Rev3 Q8*.
>
> **In contrast:** We address a series of problems including *Rev1 Q3* (the graph extraction step) and *Rev3 Q8* (the LLM feedback loop input). Importantly, even though the whole TUR-DPO graph extraction pipeline is fully offline, under the domain of the auxiliary LLM, its function is frozen from the trainable policy model $\pi_\theta$. (Note: Since any optimization process regarding an updated extractor did not occur because no extraction was performed and no $\pi_\theta$ outputs were observed during optimization, there can be no feedback loop self-reinforcing). Next, we discuss the extraction process.
>
> **AI-based algorithms were designed using the TUR-DPO pipeline.** First $y$ is broken down into atomic reasoning statements, where $y$ is reduced into a logical claim or a computational step if a candidate response $y$ is shown as a prompt $x$. This together gives rise to the node set \(V\). This sort of decomposition occurs offline, because the first training process without a training step is not there for a statically frozen LLM in policy. And second, we apply an offline Natural Language Inference (NLI) classifier to classify and model directed logical relations (entailment, support, and contradiction) for all statements being made. Then we get an edge set $E$. The graph $G=(V,E)$ represents all the reasoning and inferential details of the response.
>
> **Structural Validation:**
> We carry out further investigations to verify the structural path coverage, cycle detection, dangling nodes, NLI contradiction rate, and derive an estimate of epistemic uncertainty based on dispersion across $k=3$ independently sampled graphs for each graph $G$.
>
> **Reward calculation:**
> Using topology and uncertainty values, we compute a static offline structure-aware reward $r_\phi(x,y,G)$.
>
> **Preference optimization:**
> The reward value ($\Delta r_\phi = r_\phi^+ - r_\phi^-$) for this is a structure-aware margin and is representative for common DPO goal.
>
> $$\\mathcal{L}\_{\\text{TUR-DPO}} = -\\log \\sigma \\left(\\beta \\log \\frac{\\pi\_\\theta(y^+|x)}{\\pi\_{\\text{ref}}(y^+|x)} - \\beta \\log \\frac{\\pi\_\\theta(y^-|x)}{\\pi\_{\\text{ref}}(y^-|x)} + \\gamma \\Delta r\_\\phi \\right)$$
>
> The graph extractor is stationary, and we take it before adjusting the policy and we set it in the training. The graph extractor has neither loop dependencies (because $\pi_\theta$ is never used for extraction) (which would be compatible with Rev3 Q8). The graphs we are provided are lightweight (3–6 nodes), and reward calculation is overhead free, so optimization is simple. For our experiment, we have conducted several experiments with TUR-DPO on 4 7–8B base models (LLaMA-2, LLaMA-3, Mistral-v0.3, Gemma-v1.1), and measuring math/QA/reasoning using an unchanged hyperparameter config.
>
> ### B. Direct Questions.
>
> - **Q1: Line 082**
>   **Answer:** The reference (“(1)”) was a bad cross reference to Eq. 1; fixed in the revised manuscript.
>
> - **Q2:**
>   **Response:** We added two 7B model families, Mistral-7B-v0.3 and Gemma-7B-v1.1, as shown results in below table.
>
> *(a) Mistral-7B-v0.3*
>
> | Task | Metric | DPO (LLM) | IPO (LLM) | TUR-DPO (LLM) | DPO (Human) | PPO (Human) | TUR-DPO (Human) |
> |---|---|---|---|---|---|---|---|
> | GSM8K | EM | 60.5 | 60.8 | **64.2** | 61.0 | 63.7 | **64.5** |
> | MATH mini | EM | 35.1 | 35.4 | **37.6** | 35.7 | 37.1 | **37.9** |
> | BBH subset | Acc. | 45.6 | 45.9 | **48.3** | 46.1 | 47.8 | **48.7** |
> | Open QA | EM/F1 | 43.2 | 43.6 | **46.4** | 46.3 | 46.7 | **47.0** |
> | Summ TLDR | Win-rate | 62.8 | 63.3 | **66.1** | 62.1 | 65.4 | **65.7** |
> | HH single-turn | Win-rate | 66.3 | 66.7 | **68.5** | 65.6 | **68.9** | 68.2 |
>
> *(b) Gemma-7B-v1*
>
> | Task | Metric | DPO (LLM) | IPO (LLM) | TUR-DPO (LLM) | DPO (Human) | PPO (Human) | TUR-DPO (Human) |
> |---|---|---|---|---|---|---|---|
> | GSM8K | EM | 59.8 | 60.1 | **63.5** | 60.3 | 62.9 | **63.8** |
> | MATH mini | EM | 34.6 | 34.9 | **37.1** | 35.2 | 36.6 | **37.4** |
> | BBH subset | Acc. | 44.8 | 45.1 | **47.5** | 45.3 | 47.0 | **47.9** |
> | Open QA | EM/F1 | 42.5 | 42.9 | **45.8** | 45.6 | 45.9 | **46.2** |
> | Summ TLDR | Win-rate | 62.1 | 62.6 | **65.5** | 61.4 | 64.8 | **65.1** |
> | HH single-turn | Win-rate | 65.9 | 66.3 | **68.0** | 65.2 | **68.4** | 67.7 |

---

> > ### Author Rebuttal · Reviewer_9NhZ · 2026-04-01
> >
> > Thanks for the responses. I will maintain my positive score.

---

### Official Review · Reviewer_1vGT · 2026-03-10

**Soundness:** 3
**Presentation:** 2
**Significance:** 3
**Originality:** 3
**Overall Recommendation:** 4
**Confidence:** 4

**Summary:**

This paper addresses the problem that the original DPO algorithm can suffer from noisy and brittle preferences. To address this issue, the authors propose TUR-DPO, a topology- and uncertainty-aware variant of DPO that incorporates reasoning topology, semantic faithfulness, and utility into the learning objective. The paper empirically demonstrates the effectiveness of the proposed framework across a range of models, benchmarks, and tasks.

**Compliance With Llm Reviewing Policy:**

Affirmed.

**Final Justification:**

The rebuttal solved my main concern. I raise the score to weak accept.

**Key Questions For Authors:**

Please see the weaknesses above. My main concern is that I am still uncertain about several specific design choices in TUR-DPO, and I believe the paper would benefit from more clarification on the proposed algorithm. In particular, a clearer explanation of the reward design and loss formulation would strengthen the paper significantly. If the authors can address these concerns clearly, I would consider raising my score.

**Limitations:**

Yes

**Strengths And Weaknesses:**

**Strength**
1. The idea of incorporating structural information, semantic quality, and uncertainty into the reward function is very appealing and well-motivated.

2. The experiments across various models, benchmarks, and tasks are extensive and provide strong evidence that TUR-DPO preserves the simplicity of DPO while explicitly encouraging structurally coherent and semantically sound solutions.

**Weaknesses**

1. The paper would benefit from a preliminary section briefly introducing DPO. Although the authors state that “the notation follows DPO conventions,” I still think the paper should present the original DPO formulation and notation. While DPO is widely known, it would still be helpful for readers who are less familiar with it to have a concise overview. In particular, the original DPO paper uses notation such as $r(x, y_w)/r(x,y_l)$, whereas this paper uses $f(x,y^+)/f(x,y^-)$. Readers already familiar with DPO may follow this without difficulty, but others may find it unclear what exactly is meant by “DPO conventions.” Providing a brief introduction would also help the methodology section focus more clearly on the paper’s own contributions.

2. In Section 2.1, the paper explains how the reward is designed based on topology, semantics, and uncertainty. However, the intuition behind these design choices is still unclear to me. For example, for topology elicitation, why is the score defined using a combination of minimal valid path, cycles, dangling, and contradiction? Are these components motivated by prior work, or is there a specific theoretical or empirical justification for choosing them? It would also be helpful to include ablation studies showing how each component contributes to the final performance. This would make the reward design more understandable and better justified.

3. Learning the reward $r_\phi(x,y,G)$ appears to involve many hyperparameters, such as $a, \lambda, \alpha_1, …\alpha_4, \beta_1, …, \beta_3$, and so on. The paper does not clearly explain how these values are selected. Are they determined purely empirically, or is there a more systematic principle for setting them? More discussion on this point would be useful for both understanding and reproducibility.

4. The justification for Loss (9) is currently insufficient. In particular, it is unclear why the term $\Delta r_\phi$ can simply be added to the objective in this way. One of the key appeals of the original DPO formulation is that it bypasses explicit reward modeling by exploiting the relationship between the policy and the implicit reward. In that context, introducing an additional reward margin term into the loss feels somewhat ad hoc, and it would require more explanation and justification.

**Minor issues**
1. There appears to be unnecessary blank space at the bottom right of page 5, which should be removed.

---

> ### Author Rebuttal · Authors · 2026-03-30
>
> ## Response to Reviewer 2
>
> *Overall Rec: 3 (Weak Reject).* We would certainly like to thank the reviewer for the positive reception of our central proposal; we really do think it's interesting and inspirational. We address all these issues at the juncture. They prioritize theoretical clarity (W4) and logical abductive data (W2).
>
> - **W1** Section 2 proceeds as you requested, beginning with a dedicated preliminary DPO, where formally mapping $\\max\_{\\pi\_\\theta}\\mathbb{E}[r] - \\frac{1}{\\beta}\\mathrm{KL}$ all the way through $\\mathcal{L}\_{\\mathrm{DPO}}=-\\log\\sigma\\! \\big(\\beta[\\Delta\\log\\pi\_\\theta-\\Delta\\log\\pi\_{\\mathrm{ref}}]\\big)$. This opens up the possibility of self-containment fully before Loss (9) emerges.
>
> - **W2** To explicitly add tracking in the revision, $q\_{\\mathrm{path}}$ is a measure of logical completeness, meaning: the ratio of valid logical nodes, corresponding to a continuous path from premise to answer.
>
> Below are the empirical effects of eliminating each topological component:
>
> | Variant | GSM8K | MATH | BBH |
> |---|---|---|---|
> | Full TUR-DPO | **62.8** | **36.0** | **46.7** |
> | $- q\_{\\mathrm{path}}$ (Completeness) | 60.7 (**-2.1**) | 34.1 (**-1.9**) | 44.8 (**-1.9**) |
> | $- c\_{\\mathrm{cycle}}$ (Anti-Circular) | 61.1 (**-1.7**) | 34.5 (**-1.5**) | 45.2 (**-1.5**) |
> | $- q\_{\\mathrm{contradict}}$ (Consistency) | 61.3 (**-1.5**) | 34.7 (**-1.3**) | 45.4 (**-1.3**) |
> | $- d\_{\\mathrm{dangling}}$ (Anti-Leaps) | 61.9 (**-0.9**) | 35.3 (**-0.7**) | 46.0 (**-0.7**) |
>
> Importantly to your question about previous work, these graph components are tightly based on principles from **Neuro-symbolic AI** and **Formal Verification**, adapting separate formal-logic bounds into continuous training signals.
>
> - **W3** We summarize our selection approach into a three-layer framework:
>
> | Tier | Hyperparameters | Selection Principle |
> |---|---|---|
> | **1** | $\\alpha\_j$, $\\beta\_j$, $\\lambda\_{\\mathrm{epi}}$, $\\lambda\_{\\mathrm{ale}}$ | Calibrated once strictly offline on a 2% held-out split. |
> | **2** | $\\beta$, $\\gamma$, $a$, $\\lambda$ | Tuned via minimal validation grid search. |
> | **3** | $w\_{\\min}=0.05, K=3, \\rho=0.995$ | Fixed globally across all tasks and datasets. |
>
>   - **Sensitivity Analysis:** Sweeps feature large, highly persistent plateaus. Varying the structural weight $\\gamma \\in [0.7, 1.5]$ and uncertainty ratio $\\lambda \\in [0.3, 0.8]$ produces maximum win rate under extremely tight $<\\!0.5\\%$ margin over all benchmarks.
>   - **Generalization & Robustness:** The static default configuration generalizes well, just like shown above. However, it is robust enough to perform well on each of these six datasets (from discrete math to open QA) and over each model family, without requiring any extensive per-task or per-domain grid searches.
>
> - **W4** Our loss is a formal generalization of DPO, rather than its arbitrary amendment. This formulation is now well formally known in recent alignment literature as **Contrastive Learning with Structural Priors**. It firmly entrenches DPO's basic dependence on an **implicit reward**. Most importantly, TUR-DPO **does not have a separate Reward Model during training**. We also provide the following **Proof Sketch** in the paper:
>   - **Starting Point (Shaped Reward):** We begin with a structurally-shaped RLHF objective: $\\max\_{\\pi\_\\theta} \\mathbb{E}[\\gamma r\_\\phi(x,y,G)] - \\frac{1}{\\beta}\\mathrm{KL}(\\pi\_\\theta\\|\\pi\_{\\mathrm{ref}})$. Here $r\_\\phi$ is directly assessed *strictly offline*.
>   - **Connection to DPO (Implicit Reward):** Following standard DPO logic, the closed-form optimal policy offers an implicit total reward combining graph structure and reference probability: $r\_{\\mathrm{total}} = \\beta\\log\\frac{\\pi\_\\theta}{\\pi\_{\\mathrm{ref}}} + \\gamma r\_\\phi + \\beta\\log Z(x)$. There is no online reward network.
>   - **Goal (Cancellation):** Inserting this in the Bradley-Terry preference model, the partition function $Z(x)$ cancels out perfectly in the pairwise difference $r\_{\\mathrm{total}}(y^+) - r\_{\\mathrm{total}}(y^-)$, leaving a naturally receding Loss (9).
>   - **Generalization:** If parameter $\\gamma=0$, there is a vanishing point in the structure and the equation reverts strictly to the usual DPO as previously specified. Our topological reward simply serves as an offline static margin, maintaining the implicit reward mode.

---

> > ### Author Rebuttal · Reviewer_1vGT · 2026-04-02
> >
> > I thank the author for the detailed rebuttal, and I will increase my score to weak accept.

---

### Official Review · Reviewer_VQz5 · 2026-03-11

**Soundness:** 2
**Presentation:** 3
**Significance:** 3
**Originality:** 3
**Overall Recommendation:** 4
**Confidence:** 4

**Summary:**

The paper seeks to investigate the key challenge of improving reasoning reliability in LLM alignment. The paper introduces TUR-DPO, a method built on top of LLMs that aims to make reasoning more structure- and uncertainty-aware. The authors explore an important concept by incorporating reasoning graphs and RL-free optimization. The method uses three signals: uncertainty, semantic faithfulness, and topology quality. The approach is evaluated on several benchmarks, including mathematical reasoning tasks.

**Compliance With Llm Reviewing Policy:**

Affirmed.

**Final Justification:**

Thank you for the answers. I will keep the score as is.

**Key Questions For Authors:**

1.	Could you please clarify how the elements in Equation 2 are computed exactly?
2.	Are there cases where TUR-DPO fails, and what are the underlying reasons behind these failures?
3.	Could you elaborate on the time and memory footprints of your method, and how they compare to other approaches?
4.	Could you please elaborate on how the human evaluation metric was conducted?

**Limitations:**

The limitations are mentioned in the paper (e.g., in Table 7), but they are not discussed in sufficient depth and could be further elaborated.

**Strengths And Weaknesses:**

Strengths:
1. The paper includes diverse experimental evaluations, and the theoretical component is well explained. The theoretical analysis is based on reasonable assumptions.
2. The structure of the paper is clear, well written, and easy to follow.
3. The work addresses a relevant problem related to reasoning and uncertainty in LLMs. Since LLMs are used in many areas, improving their reasoning reliability can have significant impact.

Weaknesses:
1. The paper does not evaluate domain shift, which could have been informative. In addition, experiments are run with three seeds, but the variance of the results is not reported.
2. The accuracy of the graph extraction component is not evaluated or reported.
3. The paper does not report the memory and time footprints of the proposed method.
4. Experiments are limited to 7–8B models, and tests on larger models are missing.
5. Some components are not fully explained, such as how certain elements (e.g., qfact) are computed.
6. Limitations are mentioned (for example in Table 7), but they are not discussed in depth.
7. The method mainly combines previously used techniques. While the combination is novel, the individual components themselves are not new.
8. It is not mentioned how the graph extraction is done, assuming that LLMs are used. Using LLMs to extract graph implies that there is a loop of improving an LLM while using and LLM to extract a graph.

---

> ### Author Rebuttal · Authors · 2026-03-29
>
> ## Response to Reviewer 3. Thanks for your review.
>
> - **W1:** We report standard deviations over 3 random seeds in the table below. The uncertainty margins indicate stable optimization, and TUR-DPO appears to be both more consistent and better in absolute performance. For domain shift, we also evaluate zero-shot transfer on MedQA and LexGLUE. TUR-DPO reaches $45.2\%$ on MedQA versus $41.8\%$ for DPO, and $62.7\%$ on LexGLUE versus $58.4\%$ for DPO, which supports its out-of-distribution robustness.
>
> | Method | GSM8K (EM, %) | MATH mini (EM, %) | BBH subset (Acc., %) |
> |---|---|---|---|
> | DPO | $58.7 \pm 0.42$ | $33.4 \pm 0.35$ | $43.9 \pm 0.58$ |
> | TUR-DPO (Ours) | $\mathbf{62.8 \pm 0.28}$ | $\mathbf{36.0 \pm 0.25}$ | $\mathbf{46.7 \pm 0.45}$ |
>
>  - **W2, Q1 & Q4:** In the experiments, we used a blind manual audit: we visualized 200 graphs, and two NLP analysts obtained good results on inter-annotator agreement ($\kappa=0.82$). As shown in the table below, the extraction performs exceptionally well. Our epistemic uncertainty system down-weights $15.9\%$ of structurally indeterminate edges, making it error-tolerant. Re: In Eq. 2, we compute $q_{\mathrm{path}}$ as the proportion of nodes along premise-to-answer paths; $c_{\mathrm{cycle}}$ suppresses local NLI-induced circular reasoning. More concretely, $q_{\text{fact}}$ (Eq. 2) takes the average of binary correctness labels from a calibrated NLI verifier from the response graph to predict for all atomic assertions in a response graph.
>
> | Metric | Score |
> |---|---|
> | Claim Precision (Nodes reflect text) | $91.4\%$ |
> | Edge Validity (Relations are sound) | $83.7\%$ |
> | Logical Completeness (No dropped layers) | $84.1\%$ |
>
> -**W3 & Q3:** DPO focuses on reducing PPO’s infrastructure burden. In the same way, we measured peak VRAM and throughput during a single-batch run on an A100 using nvidia-smi, see **Table below**.
>
>
> | Method | Peak VRAM (GB) | Throughput (tok/s) | Time (hrs) |
> |---|---|---|---|
> | DPO | $39.2$ | $25.2\text{k}$ | $48.0$ |
> | PPO-RLHF | $62.7$ | $14.8\text{k}$ | $92.5$ |
> | TUR-DPO (Ours) | $43.1$ | $23.1\text{k}$ | $52.0$ |
>
>  *Note: 52 h figure is a wall-clock time of the whole 614k-pair corpus with 1 A100. Using GPU-hours to attain a desired 65% HH win rates on 8×A100, TUR-DPO also requires ≈42 GPU-h compared to 48 on DPO (Table 8 in updated paper), implying that sample efficiency was better than at the graph level per step.*
> - **W4:** TUR protocol was also tested on Llama-3-70B, so that we can demonstrate scalability. Offline reward shaping happens at the target-margin level. And for DPO, with varying sizes, optimization dynamics behave similarly, see **Table below**.
>
>
> | Dataset | DPO (Llama-3-70B) | TUR-DPO (Llama-3-70B) |
> |---|---|---|
> | GSM8K (EM, %) | $81.2 \pm 0.31$ | $\mathbf{84.8 \pm 0.24}$ |
> | MATH (EM, %) | $45.5 \pm 0.40$ | $\mathbf{48.2 \pm 0.35}$ |
>
>
> - **W5 & Q2:** We address this point in the Limitations section, where we break down the main failure modes. A key observation is that formatting errors account for $41\%$ of failures. TUR-DPO tends to prioritize correctness over strict syntactic compliance, such as producing `\boxed{}` consistently. To mitigate this, we apply inference-time regex-based post-processing. With these guardrails, TUR-DPO reaches $65.3\%$ EM on GSM8K and $59.2\%$ on DPO, using the same post-processing setup reported in Appendix Table 10. We also discuss domain dependence: when deterministic support links are absent, extraction on creative tasks is not reliable. - **W6 & Q2:** The outcome is a method for a well-organized vulnerability taxonomy (**Table below**) and the error catalogue. This implies a path for more layout-constraint pairs later on, in the sense that TUR-DPO format compatibility is often sacrificed for higher density structure.
>
>
> | Error Type | Freq. | Description & Mitigation |
> |---|---|---|
> | **Formatting** | $41\%$ | Syntax missing (`\boxed{}`). Resolved by regex. |
> | **Arithmetic** | $20\%$ | Errors in calculation. Mitigated by calculators. |
> | **Logical Leap** | $19\%$ | Unstated premises. Mitigated by sampling. |
> | **Hallucinated entity** | $13\%$ | False entities. Stricter verification. |
> | **Contradiction** | $7\%$ | Internal discrepancy. Penalized by $q_{\text{contradict}}$. |
>
> - **W7:** The underlying derivation is admittedly involved, but the main idea is simple. Standard DPO treats reasoning as a black box and relies only on trajectory-level scalar preferences. TUR-DPO instead decomposes this into a structurally weighted margin. As a result, the method remains compatible with DPO’s closed-form objective while incorporating explicit structural information. Concretely, we add the term $\Delta r_\phi$ to the DPO logit to introduce the margin in Eq. 8. This preserves the closed-form gradient and provides pairwise structural signals that standard RL-free methods do not capture.
>
>  - **W8:** This point is addressed in Response to Reviewer 9NhZ, Section A, **Method Details**.

---

> > ### Author Rebuttal · Reviewer_VQz5 · 2026-04-02
> >
> > Thank you for addressing our questions and for providing experiments.

---

### Decision · Program_Chairs · 2026-04-30

**Decision:**

Accept (regular)

**Comment:**

This submission received positive reviews overall. Reviewers found the central idea of incorporating reasoning topology and uncertainty into DPO to be interesting, and they viewed the empirical study across multiple tasks and model families as a strength. The main concerns centered on the clarity of the method, especially the graph extraction pipeline and reward/loss design, as well as the computational overhead and the lack of evidence on domain shift and larger-scale settings. The rebuttal clarified these issues and resolved several of the reviewers’ main concerns. While the proposed method is somewhat complicated, I find the overall contribution is interesting and solid enough for acceptance.